# Extensive modulation of the circulating blood proteome by hormonal contraceptive use across two population studies

Nikola Dordevic [1,12], Clemens Dierks [2,3,12], Essi Hantikainen[1], Vadim Farztdinov[4], Fatma Amari[4], Vinicius Verri Hernandes[1,11], Alessandro De Grandi[1], Francisco S. Domingues[1], Orr Shomroni[4], Kathrin Textoris-Taube[4], Vivien Bahr [5], Hannah Schmid [5], Ilja Demuth [5,6], Florian Kurth[2,7], Michael Mülleder [4], Peter Paul Pramstaller [1,8], Johannes Rainer [1] ✉ & Markus Ralser [2,9,10]

## Abstract

**Background** The study of circulating blood proteins in population cohorts offers new avenues to explore lifestyle-related and genetic influences describing and shaping human health.

**Methods** Utilizing high-throughput mass spectrometry, we quantified 148 highly abundant proteins, functioning in the innate and adaptive immune system, coagulation and nutrient transport in 3632 blood plasma, and 500 serum samples from the CHRIS and BASE-II cross-sectional population studies, respectively. Through multiple regression analyses, we aimed to identify the main factors influencing the circulating proteome at population level.

**Results** Many demographic covariates and common medications affect the concentration of high-abundant plasma proteins, but the most significant changes are linked to the use of hormonal contraceptives (HCU). HCU particularly alters amongst others the levels of Angiotensinogen and Transcortin. We robustly replicated these findings in the BASE-II cohort. Furthermore, our results indicate that combined hormonal contraceptives with ethinylestradiol have a stronger effect compared to bioidentical estrogens. Our analysis detects no lasting impact of hormonal contraceptives on the plasma proteome.

**Conclusions** HCU is the dominant factor reshaping the high-abundant circulating blood proteome in two population studies. Given the high prevalence of HCU among young women, it is essential to account for this treatment in human proteome studies to avoid misinterpreting its impact as sex- or age-related effects. Although we did not investigate the influence of HCU-induced proteomic changes on human health, our data suggest that future studies on this topic are warranted.

## Plain language summary

Millions of women use hormonal contraceptives which can cause side effects, such as skin issues, stomach problems, mood changes, and high blood pressure. In two population studies, we studied the effects of age, sex, body mass index and hormonal contraceptives on the abundance of proteins that are necessary for all aspects of normal body function. We found that hormonal contraceptives had by far the biggest impact on over one-third of the examined proteins, many of which are related to health status and lifestyle. Contraceptives with synthetic estrogen had a stronger effect than those made to be chemically identical to the ones naturally occurring. However, we found no lasting changes in blood proteins after stopping contraceptive use. These results highlight the importance of considering contraceptive use in future research to distinguish the role of contraceptives from age and sex effects, and to better understand their impact on health.

The onset, progression and outcome of human disease are affected by demographic covariates, including age, sex, and BMI, but also environmental and intrinsic variables, such as disease history or metabolism. However, in many cases the mechanisms at play are only insufficiently investigated, and we are only at the beginning to adjust treatments and their procedures according to such covariates. The Cooperative Health Research in South Tyrol (CHRIS) study[1] is a single-site population-based study aimed to investigate the genetic and molecular basis of common age-related chronic conditions and their interaction with lifestyle and environment. In recent work, we have been evaluating the impact of age, sex, and diet, amongst others, on the human metabolome[2]. Moreover, gene-metabolite associations[3], as well as genetic and metabolomic

determinants of disease[4] and age-related morbidity markers[5] have been investigated.

Recently, the human proteome has shifted to the center stage for detection and analysis of disease modifying covariates. The abundance of protein biomarkers, their respective isoforms, potential post-translational modifications, and protein sequence variants provide a snapshot of the current physiological state of the circulatory system and all organs that blood interacts with[6–9]. In population studies, large-scale plasma proteomics increasingly provides opportunities to study non-genetic associations to health-related traits, such as markers for lifestyle and environmental exposure or to detect and characterize onset and progression of disease through longitudinal monitoring of protein abundance changes[10–13]. The quantification of the plasma proteome is however a formidable challenge due to a combination of factors: the exceptionally high abundance of selected plasma proteins, the wide dynamic range of protein concentrations, the substantial sequence variability of certain proteins, and the fluctuations in protein abundances in response to diseases, physiological changes, or lifestyle factors. This challenge is particularly pronounced in large-scale studies, where it becomes even more daunting due to the heightened technical complexities involved[7,12].

Different technologies emerged to measure proteins in human blood plasma or serum samples. These range from optimized single-protein assays to targeted and untargeted mass-spectrometry (MS) based workflows to affinity-based multiplex assays[10,12,14]. Herein, we exploited the high specificity of mass spectrometry in the quantification of highly abundant plasma proteins in neat plasma. By functioning in nutrient transport, coagulation, and immune system activity, the highly abundant plasma proteins are of particular importance to understand disease biology and are attractive for biomarker discovery and assay development[15,16].

To efficiently measure these proteins in neat plasma, we recently introduced a new platform technology that combines a semi-automated sample preparation workflow and analytical flow rate chromatography for gradient lengths of 0.5–5 min. It is specifically optimized for data-independent acquisition (DIA)-MS acquisition schemes, and an in-house developed data processing software suite which integrates artificial neural networks in raw data processing (DIA-NN)[17–19]. This platform achieves high measurement precision and is highly cost-effective in the processing of large plasma proteome studies, where it provides robustness, and quantification consistency. Using Scanning SWATH[20] acquisition on Triple TOF 6600 instruments (Sciex) we herein create an MS-based plasma proteomics data set with low technical variability for $n = 3632$ CHRIS participants. To characterize this plasma proteome profile at population scale and identify the main influence factors, we perform an unsupervised, global exploratory analysis followed by specific multiple regression-based analyses testing for associations with demographic factors as well as the impact of common medications on the concentrations of circulating proteins. While all of them left signatures on the plasma proteome, we find hormonal contraceptives to be the main factor explaining the variation in human plasma protein abundance. We validate these findings on serum proteomics data from an independent cohort, the Berlin Aging Study II (BASE-II)[21–23].

## Methods
### Cooperative Health Research in South Tyrol (CHRIS) Study
In CHRIS, study participants were recruited from the adult (≥18 years) population of the middle and upper Vinschgau/Val Venosta district located in the mountainous northern-most region of Italy[1]. Next to collection and subsequent biobanking of blood and urine samples a self-reported, questionnaire-based health assessment was performed. Medication information was collected by scanning the barcode of the medication boxes study participants brought along and assignment of the respective Anatomical Therapeutic Chemical (ATC) codes. Standard blood parameters were measured in blood samples at the Hospital of Merano using standardized clinical assays. Details on measurements of clinical laboratory parameters for the present study set including the description of sample handling are described in ref. 24. In brief, antithrombin was measured in plasma citrate

samples using the enzymatic Siemens Innovance Antithrombin assay on a SYSMEX CA1500 system (Roche) and for a subset of participants using the STA-Stachrom AT III assay on a STA COMPACT MAX instrument (Stago). Albumin, high-density lipoproteins (HDL), low-density lipoproteins (LDL), triglycerides and transferrin were measured in serum samples using the colorimetric ALB plus Albumin BCG assay (Cobas), the enzymatic HDL-C Plus 3 generation assay (Cobas), the enzymatic LDL-C plus 2nd generation assay (Cobas), the Triglyceride GPO-PAP assay (Cobas) and the immunological Tina-quant Transferrin ver.2 assay (Cobas), respectively, on a MODULAR PPE (Roche). For a subset of samples, the colorimetric ALBUMIN BCG assay, the ULTRA HDL assay, the DIRECT LDL assay, the TRIGLYCERIDE assay, and the immunological TRANSFERRIN assay, respectively, were used on an ARCHITECT instrument (Abbot diagnostics). Hemoglobin (HGB) was measured in EDTA plasma using the electronic impedance laser light scattering based assay on a CD SAPPHIRE instrument (Abbot diagnostics) and on a subset of samples on an SYSMEX XN-1000 (Roche).

Of the 13,393 participants of the CHRIS study, 3632, participating between August 2011 and August 2014, were selected for mass spectrometry-based quantification of their plasma proteome.

### Sample preparation
In this study, a total of 5125 samples were subjected to proteomic measurement. Among these, 479 samples were quality control samples, 498 were standardized, commercially available plasma samples and 200 were pooled study samples used to monitor measurement quality and control technical variation. The measurement consisted of 3948 CHRIS study samples including 350 samples from a CHRIS substudy[25], which were, however, excluded from the present data analysis. Plasma citrate samples were randomly distributed across fifty 96-well plates, together with 4 study pools, 4 standardized serum (SER-SPL, ZenBio) samples, and 8 standardized plasma samples (HSER-P500ML, ZenBio) per plate.

Semi-automated in-solution digestion was performed as previously described for high-throughput plasma proteomics[18]. All stocks and stock plates were prepared in advance to reduce variability and were stored at −80 °C until use. Briefly, 5 μl of thawed samples were transferred to the denaturation and reduction solution (50 μl 8 M urea, 100 mM ammonium bicarbonate (ABC), 5 μl 50 mM dithiothreitol per well) mixed and incubated at 30 °C for 60 min. 5 μl were then transferred from the iodoacetamide stock solution plate (100 mM) to the sample plate and incubated in the dark at room temperature for 30 min before dilution with 100 mM ABC buffer (340 μl). 220 μl of this solution was transferred to the pre-made trypsin stock solution plate (12.5 μl, 0.1 μg/μl) and incubated at 37 °C for 17 h (Benchmark Scientific Incu-Mixer MP4). The digestion was quenched by the addition of formic acid (10% v/v, 25 μl) and cleaned using C18 solid phase extraction in 96-well plates (BioPureSPE Macro 96-Well, 100 mg PROTO C18, The Nest Group). The eluent was dried under vacuum and reconstituted in 60 μl 0.1% formic acid. Insoluble particles were removed by centrifugation and the samples were transferred to a new plate.

### Liquid chromatography and mass spectrometry
Measurements were performed in 17 batches, each consisting of three 96-well plates (except for MS batch 17), over a span of 6 months. Each plate included 79 study samples, 4 replicates of the study pool, 1 procedural blank, 4 standardized serum, and 8 standardized plasma samples. These serve as quality control (QC) and reference samples to compare, cross-reference, and join large studies[26,27]. The digested peptides were separated on a 5-min high-flow rate chromatographic gradient and recorded by mass spectrometry using Scanning SWATH[20] on two Infinity II HPLC systems (Agilent) coupled with a 6600 TripleTOF instrument (SCIEX). 5 μg of the sample were injected onto a reverse phase HPLC column (Luna®Omega 1.6 μm C18 100A, 30 × 2.1 mm (Phenomenex)) and resolved by gradient elution at a flow rate of 800 μl/min and column temperature of 30 °C. All solvents were of LC-MS grade. The fast separation used 0.1% formic acid in water (Solvent A) and 0.1% formic acid in acetonitrile (Solvent B) using an alternating

column regeneration system where the gradient separation of one sample is performed on one LC column by a gradient pump while a second identical column is being washed and equilibrated using a regeneration pump. The gradient separation, wash, and equilibration programs are shown in Supplementary Table S1. For MS analysis, the scanning SWATH precursor isolation window was 10 m/z, the bin size was set to 20% of the window size, the cycle time was 0.52 s, the precursor range was set to 400–900 m/z, the fragment range to 100–1500 m/z as previously described in ref. [20]. An IonDrive TurboV source (Sciex) was used with ion source gas 1 (nebulizer gas), ion source gas 2 (heater gas), and curtain gas set to 50 psi, 40 psi and 25 psi, respectively. The source temperature and ion spray voltage were set to 450 °C and 5500 V, respectively.

### Data processing, statistics, and reproducibility
Raw MS data was processed using DIA-NN v1.8[17]. We fixed the mass accuracies and the scan window size to ensure the reproducibility of our results (MS1: 12 ppm; MS2: 20 ppm; scan window size: 6). An external, publicly available spectral library was used for all measurements[28]. The spectral library was annotated using the Human UniProt[29] isoform sequence database (Proteome ID: 3AUP000005640).

All preprocessing steps of the DIA-NN output matrix were performed in the R programming language (v4.0.4). All libraries employed were in compatible versions of the R version used. The data matrix consisted, after removal of 55 outliers and all QC samples of abundances for 6762 peptide precursors in 4093 samples. Outlier removal was based on the number of detected precursors, their amino acid count, average charge, and the number of recorded missed tryptic cleavages. Peptide features with more than 40% missing values across all study samples were excluded, reducing the data set to a final number of 2716 precursors. Imputation of missing values was performed using the knn function implemented in the impute package, with $k = 9$ nearest neighbors applied to samples within each MS batch. Data were normalized using cyclic loess (with the "fast" option)[30,31] and plate effects were corrected using the removeBatchEffect function from the limma Bioconductor package[32]. To map peptide precursors to proteins, precursor filtering (retaining only $n = 2386$ proteotypic precursors) and median polish summarization implemented in the preprocessCore Bioconductor package were applied.

Functional analysis was performed with gProfiler package[33]. Only GO Terms with a false discovery rate-corrected $p$-value < 0.05 were considered significant. Protein class information was obtained from the PANTHER Classification System[34]. The results of the functional annotations are summarized in Supplementary Data 2. To assess the predictive ability of AGT, an age-matched control group was defined using the MatchIt package with an "optimal" matching strategy and calculation of propensity scores by a generalized linear model. ROC curves were calculated using the pROC package.

To compute the coefficient of variation (CV) for each protein or peptide precursor, the empirical standard deviation (the square root of the variance) was divided by the empirical mean (the average abundance), and the result was expressed as a percentage. For principal component analysis (PCA), protein abundances were scaled to zero mean and standard deviation of one (autoscaling or z-score transformation). To identify associations with sex, age, body mass index (BMI), fasting status, and hormonal contraceptive use (HCU), linear regression models (as implemented in R base) were fitted separately for each protein using its log2 transformed abundance as response and sex, age, fasting status, BMI and hormonal contraceptive use as covariates. For easier interpretation of relative effects, participants' age was divided by 10, thus age-related coefficients and effect sizes are related to 10 years of difference[35]. For BMI, clinical categories were used[36]: underweight (category 1, BMI < 18.5), normal range (category 2, 18.5 ≤ BMI < 25), overweight (category 3, 25 ≤ BMI < 30) and obese (category 4, BMI > 30). For fasting status, a binary variable based on the self-reported fasting information from the questionnaire was used (1 for participants declaring to have had a meal within the 12 h prior to the blood draw and 0 for all others). Medication information on ATC level (as described before) was included as

a binary variable. A binary variable for oral hormonal contraceptives was defined using ATC level 3 categories "HORMONAL CONTRACEPTIVES FOR SYSTEMIC USE" (ATC3 G03A) and "ANTIANDROGENS" (ATC3 G03H). To evaluate the influence of common medications on plasma protein levels linear models with explanatory variables for age, sex, BMI, fasting status, and ATC3 or ATC4 medications were fitted to the data. Only ATC3 or ATC4 medications taken on a regular basis (at least two times per week) by at least 15 participants were considered. $P$-values from linear models were adjusted for multiple testing using the Bonferroni method. Protein associations with an adjusted $p$-value smaller than 0.05 were considered statistically significant. In addition, for categorical variables (such as sex, age, BMI, fasting, or medications), to call a protein significant, its absolute difference in log2 abundance for the variable had to be larger than its CV across the sample pools. The observed difference in concentrations is thus larger than the technical variability (calculated on the QC CHRIS Pool samples).

Categorization of female participants into groups "current hormonal contraceptive use (HCU)", "previous HCU" and "never used any hormonal contraceptives" was based on self-reported questionnaire data combined with the definition of hormonal contraceptive use described above. Females with missing or ambiguous information were excluded from the analysis. For a second analysis aimed at avoiding any potential influence from a recent pregnancy, data from women reporting a previous pregnancy was removed. To identify proteins with significant differences in abundances between these categories, linear models were fitted to the data adjusting in addition for age, BMI, and fasting status.

Additionally, linear regression models were fitted to protein concentrations standardized to mean of 0 and standard deviation of 1. Coefficients from this analysis, where differences in one unit are equal to a standard deviation of 1, are comparable and reported as "effect size". All analyses were performed on log2-transformed protein concentrations.

Hormonal contraceptive usage in the BASE-II validation cohort was also recorded by self-reported questionnaires and considered intake within the last 3 months. Given that the analysis focused exclusively on women and that all BASE-II blood samples were taken after overnight fasting, linear models were not controlled for sex and fasting status. The linear modeling and principal component analysis of the BASE-II data were conducted using identical packages as those employed for the CHRIS study dataset.

Data analysis was performed in R (version 4.2.2), R markdown documents defining and describing the analysis are available on github: https://github.com/EuracBiomedicalResearch/chris_plasma_proteome.

### Berlin Aging Study II (BASE-II)
The multi-disciplinary Berlin Aging Study II (BASE-II) aims at the identification of factors promoting a healthy aging trajectory. The BASE-II sample was recruited in the metropolitan area of Berlin, Germany. The older subgroup assessed in the medical BASE-II part consisted of 1671 participants (60–85 years) and the younger subgroup investigated here of 500 participants between 20 and 37 years of age. Please refer to the study's cohort profile for a detailed description of study design and data collection[21]. Demographics of BASE-II participants used in this study are shown in Supplementary Table S2. The proteome data of this study is presented in a parallel manuscript[37] including detailed information about data generation and processing. In brief, sample preparation of serum samples followed the same protocol as for CHRIS samples[20], with the exception that mass spectrometric measurements were conducted using a timsTOF Pro Instrument (Bruker Daltonics), operating DIA-PASEF as described previously[38,39]. Raw MS data were processed with DIA-NN, using the same spectral library also used herein (see method section for data processing and statistical analysis). Peptide quantities were normalized, batch effects corrected, missing values imputed, and peptides summarized to proteins.

### Ethics approval
CHRIS study: The study was conducted according to the guidelines of the Declaration of Helsinki and approved by the Ethics Committee of the

Health Authority of the Autonomous Province of Bolzano (Südtiroler Sanitätsbetrieb/Azienda Sanitaria dell'Alto Adige; protocol No. 21/2011, 19 April 2011). All participants gave written informed consent.

BASE-II: All participants gave written informed consent. The Ethics Committee of the Charité—Universitätsmedizin Berlin approved the study (approval number EA2/029/09). The study was conducted in accordance with the Declaration of Helsinki and was registered in the German Clinical Trials Registry as DRKS00009277.

### Reporting summary
Further information on research design is available in the Nature Portfolio Reporting Summary linked to this article.

## Results
### Study sample characteristics and general data overview
To quantify highly abundant plasma proteins in 3632 participants of the CHRIS study (demographics shown in Table 1), citrate plasma samples were randomly arrayed on 50 96-well plates assigning nuclear families to the same plate. The full set of 5125 unique samples, included 977 quality control (QC) samples, 200 pools of study samples, and 350 samples from a sub-study of CHRIS[25]. In the final dataset, the quantitative precision is estimated with a median coefficient of variation (CV) of 15.45%, representing the technical variation, and 30.97%, representing technical variation plus biological signal, for pooled controls and study samples (see Supplementary Fig. S1 for distribution of CV before and after normalization). The set of quantified proteins along with the results from the present analysis are available in Supplementary Data 1. The final data set used for this analysis consisted, after the removal of samples from pregnant female study participants, and participants with missing information for any of the traits listed in Table 1, of 148 proteins in 3472 study samples.

While for most of the proteins, the signal distribution across samples was about log-normal, some proteins, including angiotensinogen (*AGT*) or plasma protease C1 inhibitor (*SERPING1*), showed a clear bimodal signal distribution in the present data set (intensity distributions of all proteins are available as Supplementary Data 1).

### Protein coverage and variation in the CHRIS cohort
Of the consistently measured high abundant proteins, 139 were enriched in the following gene ontology biological pathways (GO:BP): complement cascade (complement activation; complement activation, classical pathway), immune response (humoral immune response; immunoglobulin mediated immune response; B cell-mediated immunity; adaptive immune response; innate immune response among others); phagocytosis, blood coagulation, hemostasis, endocytosis, response to bacterium and many others (see Supplementary Data 2). Additionally, we used the PANTHER classification system to examine the protein functionality in more detail[34]. Of the 148 plasma proteins, 134 could be assigned to 25 different functional classes. The most prominent were immunoglobulins (*n* = 27), protease inhibitors (22), serine proteases (13), components of the complement system (10), and apolipoproteins (9) (see Supplementary Data 2).

**Table 1 | Demographic characteristics of the CHRIS study participants included in the analysis**

|  | Women | Men |
|---|---|---|
| *n* | 1939 | 1533 |
| Age, mean (SD) | 45.9 (16.5) | 46.2 (16.6) |
| Not fasting, *n* (%) | 118 (6.1) | 97 (6.3) |
| Classification according to BMI, *n* (%) |  |  |
| 1: underweight (BMI < 18.5) | 41 (2.1) | 7 (0.5) |
| 2: normal (18.5 < = BMI < 25) | 1091 (56.3) | 593 (38.7) |
| 3: overweight (25 < = BMI < 30) | 518 (26.7) | 687 (44.8) |
| 4: obese (BMI > = 30) | 289 (14.9) | 246 (16.0) |

Next, we addressed the response in protein abundances across our population study, in dependency of covariates and biomedical parameters. Expressed as coefficient of variation (CV), the responsiveness of protein abundances across the study samples was on average 31.1% with the 25% and 75% quantile being 21.5% and 47.3%, respectively (CV for all proteins included in Supplementary Data 1). Among the proteins with the most stable concentration were, next to albumin (*ALB*, CV = 13.3%), 4 proteins related to blood coagulation (CV between 13.0% and 18.3%) as well as 8 proteins from the complement system (CV between 13.8% and 19.9%; see Supplementary Table S3), reflecting that most of the individuals reported no acute condition at the time of sampling. To identify highly variable proteins, we calculated a relative CV defined as the ratio between the CVs in study samples (representing combined biological and technical variance) and pooled QC samples (representing technical variance). Among the top 30 proteins with highest variance were 12 immunoglobulins (CV between 22.0% and 89.2%), 4 hemolysis-related proteins (CV between 26.9% and 54.5%) as well as the hormone transporters transcortin (*SERPINA6*, CV = 48.3%) and sex hormone binding globulin (*SHBG*, CV = 123.3%; see Supplementary Table S4). From 31 commonly used protein biomarkers, 9, including albumin (*ALB*, CV = 13.3%), antithrombin III (*SERPINC1*, CV = 13.7%) and hemopexin (*HPX*, CV = 13.8%) had stable concentrations, while 10 were highly variable with the highest CV observed for lipoprotein(a) (*LPA*, CV = 134%), *SHBG* (CV = 123.3%) and fibronectin 1 (*FN1*, CV = 88.0%; see Supplementary Table S5).

Next, we compared quantified protein abundances to clinically accredited biomarker assays. These either determine the enzymatic activity of the respective proteins or quantify protein complexes containing them (such as HDL and LDL). Many test results yielded a significant correlation (Spearman's rho) with individual protein abundance values. For instance, *APOB* and LDL levels correlated with an *R* of 0.78 and transferrin with an R of 0.68, while *APOA1* levels correlated to a somewhat lower degree with the HDL, of which *APOA1* is a component (*R* = 0.41). At the other end of the spectrum, the lowest correlation was obtained for albumin (0.32). See Supplementary Table S6 and Supplementary Fig. S2 for full results. Our data thus confirms that proteome and diagnostic tests often provide related, but complementary and sometimes divergent information, a situation that is likely caused by methods which are targeting different analytes, large lipoprotein complexes rather than an individual protein, or activities rather than abundances.

To explore the data set and to investigate the main influence factors on the present plasma proteome, we next performed a principal component analysis (PCA) on the z-score transformed abundances. This analysis revealed a subset of almost exclusively women that separated from the main bulk of study participants on principal component 1 (PC1; see Fig. 1A). A distinct set of proteins, with the strongest drivers being *AGT* and *SERPINA6*, characterize this separating subset (Fig. 1B). The same grouping was also observed in a PCA performed exclusively on data from female participants (Supplementary Fig. S3), while it was not present in a PCA on samples from male study participants (Supplementary Fig. S4). PC1, explaining the largest variance in the data set, also showed a clear relationship with the participants' age (Fig. 1C). To systematically evaluate factors that are related to this principal component, we performed a multiple regression analysis explaining PC1 by relevant covariates that included participants' age, sex, (categorical) BMI, (binary) self-reported fasting status and common medications taken on a regular basis. The strongest association of PC1 was with hormonal contraceptive-related medications, followed by age, sex, and BMI category 4 (obese; BMI > = 30) (Supplementary Table S7). In contrast, no relationship with any technical factors, such as the sample storage duration, was found. Thus, in the present data set, hormonal contraceptives are the major contributors to the variance observed on the quantified plasma proteome in a generally healthy population.

### Plasma proteome associations to sex, age, and BMI
To identify the associations of proteins with the covariates sex, age, and body mass index (BMI), we fitted multiple regression models to the abundances of

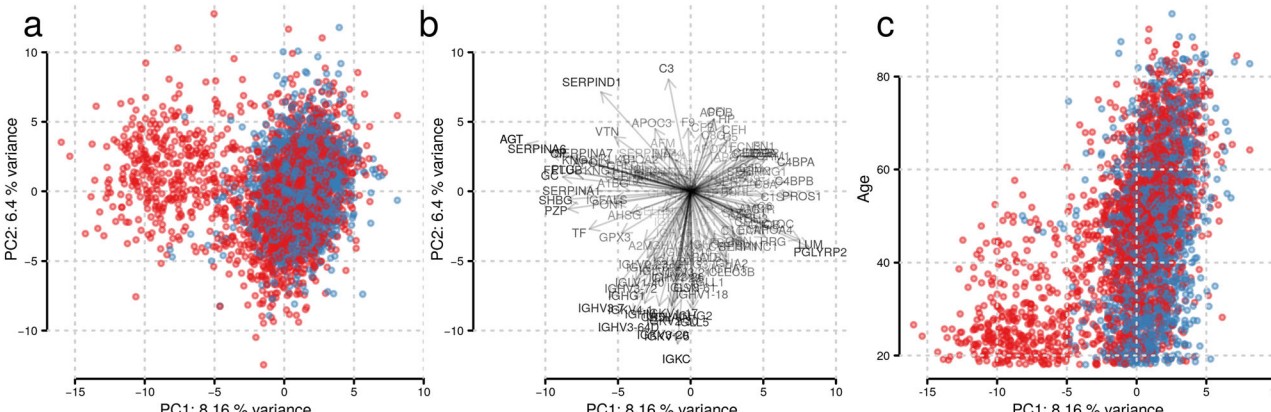

**Fig. 1 | Principal Component Analysis (PCA) of the CHRIS plasma proteome data reveals that proteomes of young women cluster into two distinct groups, which are attributable to the use of hormonal contraceptives. a** Grouping of individuals on PC1 and PC2. Women (red) and men (blue) are distinguished by color. A distinct cluster of women separates from the main bulk of male and female participants, particularly along PC1, indicating the presence of a subgroup of female study participants with a strong characteristic plasma proteome. **b** PCA loadings on PC1 and PC2. Each arrow represents a protein, with its length and direction indicating the protein's contribution to the respective principal component, highlighting the proteins most influential in driving variance. **c** Relationship between PC1 (x-axis) and participant's age (y-axis). The cluster of women on PC1 is clearly enriched with young women below the age of 40.

each quantified plasma protein. These models were additionally adjusted for the participants' fasting status and the usage of oral hormonal contraceptives due to their significant influence in the PCA (Fig. 1, Supplementary Table S7). For protein associations with categorical variables, we used more stringent significance criteria that considers also the technical noise of each individual protein: In addition to a statistical significance level (alpha = 0.05), we require the protein's observed average difference in abundances to be larger than its CV, which was determined for each protein on study-specific quality control samples measured in the same data set.

With this setting, we identified 22 plasma proteins significantly differing between female and male participants (Supplementary Table S8; Fig. 2A), with the top five candidates being retinol-binding protein 4 (*RBP4*), glycosylphosphatidylinositol specific phospholipase D1 (*GPLD1*), pigment epithelium-derived factor (*SERPINF1*), and transthyretin (*TTR*) showing lower, and ceruloplasmin (*CP*) higher abundances in women, respectively. 91 proteins were found to be significantly related to the participants' age (Supplementary Table S9; Fig. 2B), with insulin-like growth factor-binding protein complex acid labile subunit (*IGFALS*) and vitronectin (*VTN*) having the strongest effects, both showing significant negative correlations with age (see Supplementary Figs. S5 and S6). We observed multiple proteins that had significantly different concentrations when comparing participants of BMI category 1 (underweight; BMI < 18.5; $n = 48$), BMI category 3 (overweight; $25 <= BMI < 30$; $n = 1205$), BMI category 4 (obese; BMI >= 30; $n = 535$) to participants from BMI category 2 (normal; $18.5 <= BMI < 25$; $n = 1,684$). In total, we observed 4 significantly associated proteins for underweight, 4 proteins for overweight and 20 proteins for obese status (Supplementary Tables S10, S11, and S12). Most of the BMI-associated proteins showed a difference in concentrations which was consistently increasing (or decreasing) with BMI (Fig. 2C), such as *SHBG* and apolipoprotein D (*APOD*) having lower abundances with increasing BMI. Also, one protein, apolipoprotein A-IV (*APOA4*), was significantly associated with fasting status exhibiting a 9% higher abundance in non-fasting participants (Supplementary Table S13).

The effect sizes for HCU associations, described in more detail in a following section, were larger than those for sex- or age-associated proteins. Further, a large overlap in associations was observed between HCU and the other traits (Fig. 1D). Given this large impact of HCU on the plasma proteome, we evaluated to what extent adjustment for HCU influences the results of a general analysis for age, sex and BMI-associations. We thus conducted a sensitivity analysis by fitting the same linear models to the data omitting only the explanatory variable for HCU and compared the results of the two models. Indeed, 8 from the 28 sex- and 15 from the 101 age-

associated proteins identified in this sensitivity analysis were significantly related to HCU but not to sex or age in the full analysis model (see Supplementary Tables S14 and S15 for coefficients and p-values for these proteins in both analyses). In contrast, BMI-associations were not affected. Thus, the adjustment for HCU had a clear impact on the age and sex-association results while it did not affect the BMI-associations (see Fig. 3).

## Influence of Medication on the Plasma Proteome

We next determined the most common medications in the present data set and evaluated their impact on the high abundant plasma proteome. To identify associations between plasma proteins and therapeutic subgroups of general medication, we first identified all ATC level 3 medications taken by at least 15 participants (0.4% of the sample set) on a regular basis (at least two times per week) and defined a binary variable for each of them. These were then included as explanatory variables into the per-protein multiple regression models, that accounted also for age, sex, (categorical) BMI and (binary) fasting status of each study participant. Supplementary Table S16 shows the tested medications, the number of participants taking that medication, and the number of significant protein associations.

Among the 27 tested ATC3 medications, the most frequent were hormonal contraceptives for systemic use (ATC3 G03A), thyroid preparations (ATC3 H03A), antithrombotic agents (ATC3 B01A) and lipid modifying agents (ATC3 C10A), each with more than 200 participants taking these on a regular basis. In line with results from the previous section, medications related to contraception (ATC3 G03A, G03H and G02B) yielded by far the highest number of significantly associated plasma proteins (50, 38 and 19, respectively, see Supplementary Table S16). For these medications, we observed a considerable overlap of the protein signatures as well as similar effect sizes (see Supplementary Fig. S7). Even contraceptives for topical use (ATC3 G02B), which were not considered in the definition of the oral hormonal contraceptive use (HCU) variable in the previous section, showed, despite smaller effect sizes, a similar protein signature. For the remaining medications, no or only few significant protein associations were found (see Supplementary Table S16). Tables with significant proteins for each medication are provided in the supplement (Supplementary Tables S17-S32).

We subsequently repeated the analysis on ATC level 4 medications to identify additional associations with the more specific chemical or pharmacological subgroups defined by this ATC level. Also in this analysis, medications related to hormonal contraceptives yielded the highest number of significant protein associations: 52, 38, 36 and 35 for progestogens and estrogens, fixed combinations (ATC4 G03AA),

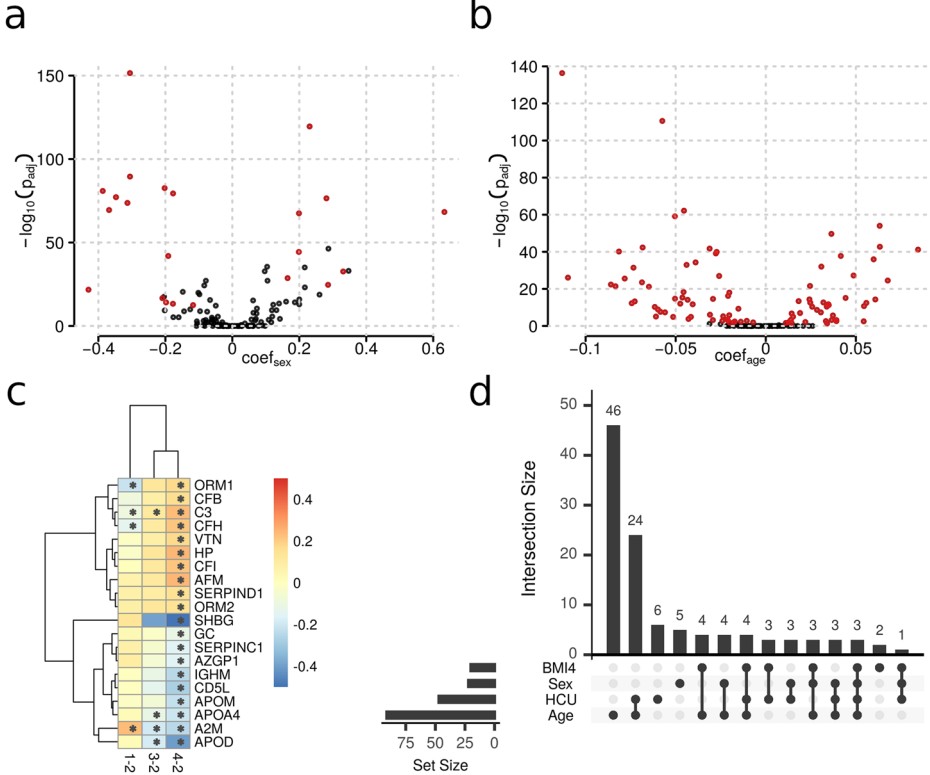

**Fig. 2 | Associations of plasma proteins with sex, age and BMI in CHRIS participants, determined by multiple linear regression analysis. a** Sex and **b** age associations represented as volcano plots. Each data point represents one protein with data points shown in a red color indicating a significant association. Analysis was performed on data from n = 3,472 study participants. The coefficient (x-axis) represents the log2 difference in average abundance between women and men and in age over 10 years difference. Bonferroni-adjusted p-values (y-axis) indicate the significance of this difference. **c** Hierarchically clustered heatmap of coefficients for proteins found to be significantly associated with at least one BMI category. The values for the coefficients are color-coded with blue colors representing negative, orange to red positive associations. Columns 1-2, 3-2 and 4-2 contain the coefficients for the comparison of BMI category 1 (underweight; BMI < 18.5), BMI category 3 (overweight; 25 <= BMI < 30) and BMI category 4 (obese; BMI >= 30) to BMI category 2 (normal; 18.5 <= BMI < 25), respectively. Significant associations are indicated with an asterisk. **d** Overlap of significant protein associations with age, sex, BMI category 4 and hormonal contraceptive use (HCU), which was included in the linear models to adjust for this medication. A large overlap of significant associations is present between HCU and any other trait.

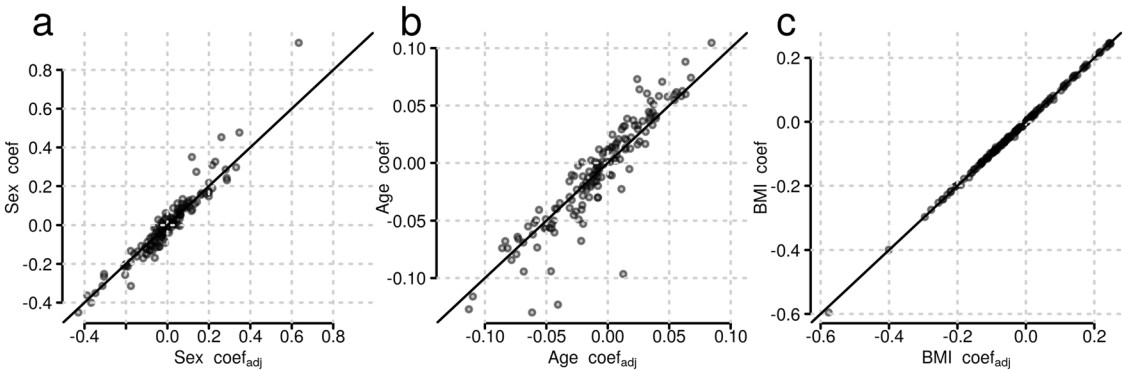

**Fig. 3 | Sensitivity analysis indicates that not accounting for hormonal contraceptive use may misattribute age and sex effects.** Shown are the coefficients from the linear model adjusting for hormonal contraceptive use (x-axis) against the coefficients from a linear model without that adjustment (y-axis) for Sex **a**, Age **b** and BMI (**c**, obese *vs* normal), respectively. The solid black line represents the identity line.

antiandrogens and estrogens (ATC4 G03HB), progestogens and estrogens, sequential preparations (ATC4 G03AB) and intravaginal contraceptives (ATC4 G02BB), respectively (see Supplementary Table S33). Further, the signatures and effect sizes were highly similar for these medications, irrespective of the route of administration (Supplementary Fig. S8): Intravaginal contraceptives had a similar signature and effect sizes than all other, orally administered, hormonal contraceptives. In contrast, no significant protein was found for the ATC4 medication intrauterine contraceptives (ATC4 G02BA), which is part of the same ATC level 3 medication subgroup (ATC3 G02B, contraceptives for topical use) as intravaginal contraceptives (ATC4 G02BB). The medication subgroup with the next most significant proteins (14) was vitamin K antagonists (ATC4 B01AA), while for platelet aggregation inhibitors excl. heparin (ATC4 B01AC, part of the same ATC3 therapeutic subgroup antithrombotic agents) only a single significant association was found. For the remaining medications, only few significant proteins were identified, and no significant association was detected for 13 of the in total 34 tested medication subgroups (see Supplementary Tables S34-S46 for significant proteins for the tested medications). Thus, summarizing, hormonal contraceptives had, among all tested medications, by far the strongest influence on the quantified plasma proteome in this study.

## Oral Hormonal Contraceptives Shape the Plasma Proteome in Female Study Participants

To directly identify the plasma proteome associated to hormonal contraceptives we next fitted multiple linear regression models to the proteomics data with explanatory variables for age, (categorical) BMI, (binary) fasting status and (binary) oral hormonal contraceptive use (HCU). To avoid any unwanted influence of age or sex on the results, we performed this analysis on the data subset of female study participants below the age of 40 (n = 729,

**Fig. 4 | Hormonal contraceptive use is strongly affecting the circulating blood proteome of women below the age of 40 in two independent cohorts.** Volcano plots illustrating the association between protein abundances and hormonal contraceptive use in the CHRIS **a** and the BASE-II **d** cohort. The sample sizes are n = 729 and n = 240, respectively. Each point represents one protein, red coloring indicates significant association. **b** and **e** Abundance of the protein angiotensinogen (*AGT*) in study participants taking hormonal contraceptives (HCU) and women and men that don't in the CHRIS **b** or the BASE-II **e** study. Samples sizes for the 3 groups are 316, 1,623, and 1,533 in CHRIS and 91, 149, and 197 in BASE-II. **c** ROC (Receiver Operating Characteristics) curve demonstrating the high predictive power of AGT for HCU. **f** Comparison of effect sizes of all proteins for association with HCU in CHRIS and BASE-II; the solid black line represents the identity line. Correlation of data points: Spearman's rho = 0.91.

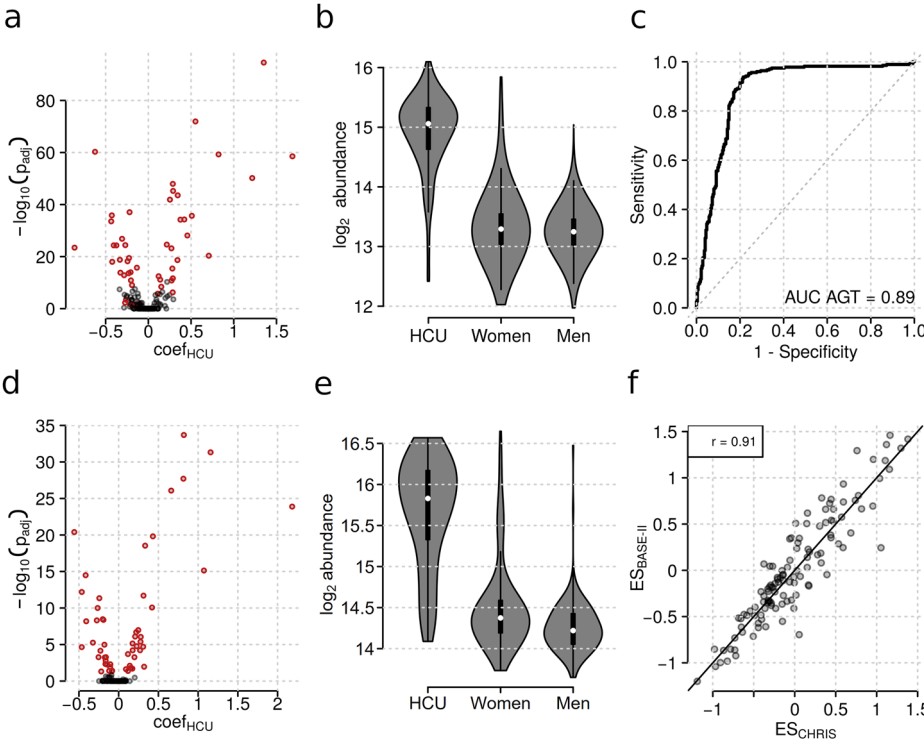

of which 275 participants reported the use of hormonal contraceptives). The results from this analysis were highly comparable to the associations for HCU from the analysis performed on the full data set (see Supplementary Fig. S9). We identified 50 plasma proteins that were significantly associated with the use of hormonal contraceptives, where the strongest association, and largest effect size, was found for angiotensinogen (*AGT*) (see Supplementary Data 3). Similarly, we observed a clear and strong difference in abundance of *AGT* between women taking oral contraceptives and all other female study participants as well as the male ones when considering the overall study population (see Fig. 4B). This large difference also explained the observed bimodal signal distribution for this protein mentioned above (see Supplementary Data 7, file 38). We further assessed the predictive power of this protein for HCU based on a sex, age, BMI, and fasting status-matched control group (n = 275) for the 275 female participants taking hormonal contraceptives. The AUROC (Area Under Receiver Operating Characteristic curve) for *AGT* was 89% confirming its high predictive ability (Fig. 4C).

### Hormonal Contraceptive Use Induces Similar Proteomic Changes in an Independent Cohort

To validate our findings, we next investigated protein associations with hormonal contraceptive use in data from an independent cohort, the Berlin Aging Study II (BASE-II)[21,23] (demographics of the study subgroup in Supplementary Table S2). A principal component analysis conducted on the serum proteome data from BASE-II participants below the age of 40, revealed a similarly strong effect of hormonal contraceptive use (see Supplementary Fig. S10). We next fitted the same linear models as used in our analysis to protein abundances of female participants of the BASE-II study (n = 240, 100 with HCU, 140 without) and derived effect sizes for the influence of hormonal contraceptives from this analysis (see Supplementary Data 4 for the full results from this analysis). Similarly strong associations for HCU were identified also in the BASE-II data (compare volcano plots for CHRIS and BASE-II in Figs. 4A and 4D) and comparable abundances of *AGT* across groups were observed (Figs. 4B and 4E). Finally, the effect sizes for HCU from the two independent studies were highly similar (Spearman's rho = 0.9; Fig. 4F) validating thus our findings.

### Combined Hormonal Contraceptives Containing Ethinylestradiol Have a Stronger Effect Than Those with Bioidentical Estrogens

Our HCU definition considers all hormonal contraceptive preparations, regardless of the actual hormone combinations. Ethinylestradiol (EE), a very potent synthetic estrogen used in most combined oral contraceptives (COC), has been shown to have a broader effect on the plasma proteome than bioidentical equivalents such as estradiol valerate[40]. To test this in our CHRIS data set, we classified the COCs into 3 main groups, COC with EE, with bioidentical estrogen (BE) or progestogen (P4) preparations and performed a linear regression analysis to evaluate their respective influence on the quantified plasma proteome. After excluding participants with non-oral contraceptives, the data set consisted of n = 412 participants without contraceptive use, n = 248 with COCs containing EE, n = 17 with COCs containing BE and n = 6 with progestogen (P4) containing preparations (see Supplementary Table S47 for the definition of the groups, numbers and respective preparations). Indeed, a higher number of significant proteins was found for EE (57 compared to 2 and 0 proteins for BE and P4, respectively), but the large difference in sample size for the different groups precludes any conclusion on p-values. However, when comparing the effect sizes of the different COCs (which are independent of the sample size), EE showed an about 3 times stronger effect on the plasma proteome than BE, while affecting about the same proteins (slope of the fitted line: 0.34, R2 = 0.46; Fig. 5A). Progesterone preparations, in contrast, seemed to affect different sets of proteins (Fig. 5B). The full results from this analysis are provided as Supplementary Data 5 and a comparison of abundances in the various categories is shown for selected proteins in Supplementary Fig. S11.

### No Long-Lasting Effects of Hormonal Contraceptives on the Plasma Proteome Observed

Recently, a long-lasting effect of menopausal hormonal therapy (MHT) in the circulating proteome was reported[41]. To test whether also hormonal contraceptives would have a long-lasting impact on the high abundant plasma proteome of the CHRIS study, we categorized female participants below 40 years of age into 3 groups: current use of hormonal contraceptives (n = 275), previous use of contraceptives (n = 280), and never used hormonal contraceptives (n = 76) and identified proteins with significant differences in

**Fig. 5 | Impact of different types of combined oral contraceptives (COCs) on the plasma proteome in CHRIS.** COCs with ethinylestradiol induce more pronounced changes in protein abundances compared to those containing bioidentical estrogen, whereas those containing progesterone affect a different set of plasma proteins. **a** Effect sizes for COCs with ethinylestradiol (n = 248) against effect sizes for COCs with bioidentical estrogen (n = 17). **b** Effect sizes of COCs with ethinylestradiol against those for progesterone preparations (n = 6). Each point represents data from one protein. Solid black lines represent the linear regression fit to the data points with its slope and p-value shown in the top left corner.

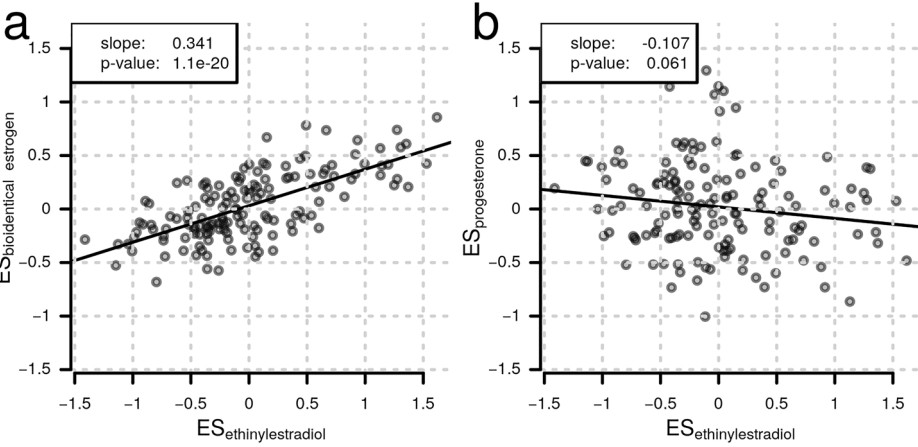

abundances between these. Restricting the analysis to young women ensured balanced groups and reduced a potential influence of age and menopause on the results. Also, all models were adjusted for age and BMI to avoid potential confounding. In contrast to current use of hormonal contraceptives, no protein was significantly changed in women with previous HCU, compared to women that never took contraceptives (Supplementary Fig. S12). To avoid any potential influence of a recent pregnancy, we repeated the analysis on data from women who declared to have never been pregnant (current use of hormonal contraceptives n = 241, previous use of contraceptives n = 178 and never used hormonal contraceptives n = 67), but the results were essentially identical (Supplementary Fig. S13). Thus, in the present data set we could not observe any long-term plasma proteome changes stemming from a previous use of hormonal contraceptives.

## Discussion

In this study, we explore the plasma proteome of 3632 CHRIS study participants. Using a combination of semi-automated sample preparation of neat plasma, fast analytical flow-rate chromatography, and Scanning SWATH[20], we processed over 5125 plasma samples (including QCs). An HPLC setup with two binary pump systems, one gradient and one wash/equilibration pump, reduced overheads to 1.8 min and eventually allowed one operator to run 3 LC-MS batches (3 × 96-well plates each) per week and to complete the measurements within only 4 weeks of instrument time. We could thus obtain precise quantities and limited batch effects, demonstrating that mass spectrometry-based proteomics suits the quantification of the high abundant plasma proteomic fraction also in large-scale epidemiological cohorts.

While the present data set is one of the larger MS-based proteomics data sets to date, ensuring adequate statistical power for the performed association analysis, it specifically addresses the high abundant plasma protein fraction, which contains many environment- and disease-responsive proteins that are of fundamental physiological relevance to humans. The high abundant plasma proteins quantified in our study are, amongst other processes, members of innate immunity such as the complement system, coagulation factors, or immunoglobulins. Furthermore, a substantial fraction of the quantified proteins are targets of FDA-approved drugs, established biomarkers, or responsive to various conditions, such as nutritional challenges, or infections[9,18,42,43]. In the CHRIS study, many of these proteins varied significantly across individuals.

We detected a high degree of concordance between our and data from other cohort studies, confirming the reliability of our proteomic dataset. For example, about half of the sex-associated proteins identified in our study, have been described in orthogonal cohorts, with the same directionality of the effect[44–51]. (see Supplementary Data 6). Also, a large portion (70%) of the age-associated proteins have been related to aging in other cohorts[48,52–58] with most of them showing a similar pattern of regulation (see Supplementary Data 6). For the newly described age associations, while not reaching

significance levels, similar trends in fold changes were reported in other cohorts as well[55,56,58]. Our dataset also confirmed several BMI-associated protein quantities that were revealed in large population studies[59,60] (Supplementary Data 6). Finally, proteins that were significantly associated with hormonal contraceptives were observed in other cohorts as well (47 out of 50)[40,61–66].

In the CHRIS cohort, the intake of hormonal contraceptives was the most dominant variable affecting the high abundant plasma proteome. In a focused analysis on data from women below the age of 40 we identified proteins strongly associated with HCU, a result we could replicate in serum proteomics data from an additional cohort, the BASE-II study. Notably, in both data sets, levels of angiotensinogen (*AGT*), separated users and non-users of hormonal contraceptives, suggesting it as a potential biomarker of contraceptive use. However, while in CHRIS *AGT* levels had the highest effect sizes, in BASE-II *SERPINA6* showed the highest effect size and lowest p-value for HCU.

Since our data is based on European cohorts, we cannot exclude the observed effects to potentially differ in the circulating proteome of other populations or ancestries. Our HCU results are consistent with prior studies that, albeit also based on data from European ancestry, studied selected proteins and their response to HCU use[62,67]. For example, HCU associations were described for angiotensinogen (*AGT*), vitamin D binding protein (*GC*), transferrin (*TF*), ceruloplasmin (*CP*), *SHBG*, transcortin (*SERPINA6*), thyroxine-binding globulin (*SERPINA7*), haptoglobin (*HP*) and Fetuin B (*FETUB*)[61–65,67,68]. Further, while this study was under consideration, a parallel study (preprint) found that oral contraceptives influence the plasma proteomes of women with untreated hypertension, identifying multiple proteins (*FETUB*, *ITIH3*, *PZP*, *PLG*, *PGLYR2*) associated with increased risk for elevated blood pressure[68]. Moreover, in a large metabolomics study[69], *ALB* was also significantly decreased with HCU, while *APOC3*, which reflects HDL levels, exhibited consistent upregulation by HCU. Despite these reports, the use of hormonal contraceptives is thus far not systematically accounted for in the typical epidemiological or clinical plasma proteome studies. A sensitivity analysis unequivocally underscored, however, the necessity of incorporating hormonal contraceptive use into the analytical models, to prevent proteins influenced by this treatment from being erroneously linked to factors such as age, sex, or correlated phenotypes. We thus conclude that the dominant effect of hormonal contraceptives might often be misconstrued as age- or sex-related effects.

Hormonal contraceptives are widely recognized for their adverse side effects on the human body. Common side effects include dermatological issues (such as acne), psychological effects (such as mood changes), neurological symptoms (like headaches), and gastrointestinal disturbances (such as nausea and bloating). More severe side effects include an increased risk of fungal infections, thromboembolisms, high blood pressure, and depression[70–75]. Despite these side effects being well-documented for a long

time, the underlying mechanisms remain underexplored. However, observed changes in the plasma proteome could significantly reflect health-related physiological changes. At least some of them might be attributed to the activity of liver cytochrome p450 (CYP) enzymes[76]. Contraceptive estrogens, in particular ethinylestradiol, are not only extensively metabolized by several of these enzymes including *CYP3A*[77], but are also known to activate respectively inhibit a large number of other CYP forms, which in turn can also lead to adverse drug-drug interactions[78]. A better understanding of the induced proteomic changes could pave the way for personalized contraceptive treatments and the development of diagnostic assays. In the liver, oral contraceptives stimulate the synthesis of steroid-binding globulins, such as *SHBG*, thereby affecting circulating, free steroid levels and they further increase low-grade inflammation, alter lipid metabolism, and affect the coagulation system, resulting in an increased risk for thromboembolic events[40]. Like oral contraceptives, and in agreement with[79], also (hormonal) intravaginal contraceptives resulted in an almost identical protein signature with highly similar effect sizes. The effect of hormonal contraceptives on the plasma proteome is thus independent of the route of administration; our analysis comparing different combined oral contraceptives suggests, however, that it may vary depending on the type of estrogen used in the respective preparations. In agreement with a small randomized controlled trial[40], our results show a stronger effect of combined hormonal contraceptives containing ethinylestradiol compared to preparations with bioidentical estrogens. Indeed, exogenous estrogens have been reported to cause upregulation of hepatic angiotensinogen[80,81] associated with an activation of the renin-angiotensin system with, however, little renal and systemic consequences[82,83]. Progestin-only contraceptives were in contrast only weakly or not at all associated with changes in blood metabolite levels[69]. Also, in our data, a weaker response of the circulating proteome to this type of contraceptive was observed, but the low number of cases ($n = 6$) prevented a conclusive answer.

Recently, a long-term effect of menopausal hormonal therapy on the circulating plasma proteome was described[41]. We could not identify any such effect for hormonal contraceptives. While this is also in line with results from a large metabolomics study[69], we would like to note, however, that from the 22 of the proteins that showed a long-term response to menopausal hormonal therapy, only alpha-1-antichymotrypsin (*SERPINA3*) was quantified in CHRIS.

To conclude, we found hormonal contraceptives to be the largest influence factor on the highly abundant plasma protein fraction with effect sizes far larger than those of any other assessed trait or medication, affecting the levels of several proteins with fundamental physiological roles, such as blood pressure control, nutrient transport, or immune system. This result was confirmed through replication of the analysis on serum proteomics data of the independent BASE-II cohort. In an analysis on a data subset, we also found, in agreement with a previous report[40], evidence of a broader effect for preparations containing ethinylestradiol, compared to those with bioidentical estrogens. We need to emphasize that we have not investigated in this study whether the changes detected in the plasma and serum proteome are related to any differential health outcome, or unwanted side effects associated with HCU. However, the impact of these medications is substantial, and we thus encourage more in-depth investigations and the design of dedicated studies, which should clarify to which degree desired and undesired effects resulting from the use of this medication are associated with the changes in the plasma proteome. Finally, due to its high prevalence, and apparently strong influence, hormonal contraceptive use should be accounted for in any epidemiological or clinical study of the plasma and serum proteome to avoid spurious findings.

## Supplementary information

Document (pdf) with Supplementary Figs. S1–S13 and Supplementary Tables S1–S47. Supplementary Data 1: spreadsheet (xlsx format) with the results from the association analyses for the CHRIS cohort. Supplementary Data 2: spreadsheet (xlsx format) with functional annotations of detected proteins. Supplementary Data 3: spreadsheet (xlsx format) with the proteins significantly associated with hormonal contraceptive use in women below the age of 40. Supplementary Data 4: spreadsheet (xlsx format) with results from the association analysis of BASE-II cohort data. Supplementary Data 5: spreadsheet (xlsx format) with the results for association between proteins and different combined oral contraceptives. Supplementary Data 6: spreadsheet (xlsx format) with the comparison of results with literature. Supplementary Data 7: zip archive containing intensity distribution plots for all proteins.

## Data availability

The mass spectrometry proteomics data for QC samples, the fasta file used for spectral library annotation, and peptide and protein quantities obtained from DIA-NN have been deposited to the ProteomeXchange Consortium (http://proteomecentral.proteomexchange.org) via the PRIDE partner repository[84]. The corresponding PRIDE identifiers for the study pools and quality controls are PXD052861 and PXD052892 respectively. For safeguarding compliance with the General Data Protection Regulations (GDPR), Italian personal data processing legislation, and the informed consent at the basis of the CHRIS study, individual-level data of study participants cannot be deposited in a public repository. However, individual-level data acquired as part of the CHRIS study data can be requested for research purposes by submitting a dedicated request to the CHRIS Access Committee. Please visit https://chrisportal.eurac.edu/ for more information on the process. A similar principle is applied for the BASE-II data. Please contact the scientific coordinator as outlined on https://www.base2.mpg.de/contact. Source data underlying the linear regression analysis results shown in Figs. 2, 4, and 5 are provided in Supplementary Data 1, 4, and 5, respectively.

## Code availability

The code to analyze the data from both cohorts is available as R markdown files in the public GitHub repository https://github.com/EuracBiomedicalResearch/chris_plasma_proteome[85]. The data analysis was performed in R (version 4.2.2) using software packages from the Bioconductor project version 3.19.

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

## Acknowledgements

The CHRIS study is a collaborative effort between the Eurac Research Institute for Biomedicine and the Healthcare System of the Autonomous Province of Bozen/Bolzano (SüdtirolerSanitätsbetrieb/Azienda Sanitaria dell'Alto Adige). The investigators thank all study participants from the middle and upper Vinschgau/Val Venosta, the general practitioners, the personnel of the Hospital of Schlanders/Silandro, the field study team, and the personnel of the CHRIS Biobank (BRIF code BRIF6107) for their support and collaboration. We thank the Charité Core Facility High Throughput Mass Spectrometry, especially Daniela Ludwig for sample preparation. We thank Dr Anita Domanegg for her feedback on the types of hormonal preparations used. The CHRIS study was funded by the Department of Innovation, Research and University of the Autonomous Province of Bolzano-South Tyrol and supported by the European Regional Development Fund (FESR1157). The authors thank the Department of Innovation, Research University and Museums of the Autonomous Province of Bozen/Bolzano for covering the Open Access publication costs. Work conducted by the Ralser lab was partially funded by Wellcome Trust (IA 200829/Z/16), the European Research Council (ERC) under grant agreement ERC-SyG-2020 951475, and the German Federal Ministry of Education as part of the National Research Node "Mass spectrometry in Systems Medicine (MSCoresys)", under grant agreement 031L0220. The funders had no role in study design, data collection and analysis, decision to publish, or preparation of the manuscript. This article uses data from the Berlin Aging Study II (BASE-II). BASE-II was supported by the German Federal Ministry of Education and Research under grant numbers #01UW0808; #16SV5536K, #16SV5537, #16SV5538, #16SV5837, #01GL1716A, and #01GL1716B. This work was supported by a grant from the Deutsche Forschungsgemeinschaft (grant number 460683900 to ID). Graphical abstract was created with Biorender.com.

## Author contributions

Conceptualization: J.R., M.R., N.D., C.D., E.H., I.D.; Investigation: V.V.H., F.A., K.T.-T., A.D., M.M., V.B., H.S., I.D.; Formal Analysis: N.D., C.D., J.R. V.F., O.S., V.B., M.M.; Writing—original draft preparation: E.H., C.D., J.R., N.D., F.A.; Writing—review and editing: N.D., C.D., E.H., M.M., V.F., V.V.H., A.D., I.D., F.S.D., H.S., V.B., F.K., P.P.P., M.R., J.R.; Funding acquisition: I.D., F.K., M.R., and P.P.P.

## Competing interests

M.R. is the founder and shareholder of Eliptica Ltd. Michael Mülleder is a consultant and shareholder of Eliptica Ltd. All other authors declare no competing interests.

## Additional information

[1]Institute for Biomedicine, Eurac Research, Bolzano, Italy. [2]Charité – Universitätsmedizin Berlin, Corporate Member of Freie Universität Berlin and Humboldt Universität zu Berlin, Institute for Biochemistry, Berlin, Germany. [3]Department of Infectious Diseases and Critical Care Medicine, Charité – Universitätsmedizin Berlin, Corporate Member of Freie Universität Berlin and Humboldt Universität zu Berlin, Berlin, Germany. [4]Charité Universitätsmedizin Berlin, Corporate Member of Freie Universität Berlin and Humboldt Universität zu Berlin Core Facility – High Throughput Mass Spectrometry, Berlin, Germany. [5]Department of Endocrinology and Metabolic Diseases (Including Division of Lipid Metabolism), Biology of Aging Working Group, Charité – Universitätsmedizin Berlin, Corporate Member of Freie Universität Berlin and Humboldt-Universität zu Berlin, Berlin, Germany. [6]Charité - Universitätsmedizin Berlin, Corporate Member of Freie Universität Berlin, Humboldt-Universität zu Berlin, and Berlin Institute of Health, Regenerative Immunology and Aging, BIH Center for Regenerative Therapies, Berlin, Germany. [7]German Center for Lung Research (DZL), Berlin, Germany. [8]Department of Neurology, General Central Hospital, Bolzano, Italy. [9]Max Planck Institute for Molecular Genetics, Berlin, Germany. [10]The Centre for Human Genetics, Nuffield Department of Medicine, University of Oxford, Oxford, UK. [11]Present address: Department of Food Chemistry and Toxicology, University of Vienna, Vienna, Austria. [12]These authors contributed equally: Nikola Dordevic, Clemens Dierks. ✉e-mail: johannes.rainer@eurac.edu

