## [Transparent Peer Review file · Communications Medicine]

Extensive Modulation of the Circulating Blood Proteome by Hormonal Contraceptive Use Across Two Population Studies

Corresponding Author: Dr Johannes Rainer

Version 0:

Reviewer comments:

Reviewer #1

(Remarks to the Author)

This study by Dordevic et al presents a useful approach to quantitative plasma proteomics, quantifying 148 proteins in a cohort of 3 632 participants using ultra-high-throughput mass spectrometry with SWATH readout. The research aligns well with the previous study introducing the concept of using SWATH with extremely short gradients, Messner et al. (2020). This has the potential to move rapid plasma profiling in clinical settings and the authors show that this technology makes it feasible to run thousands of plasma samples from a clinical cohort. However, while the concept and methodology are promising, there are critical areas where the study must be strengthened. The lack of data availability is a significant concern. For a robust peer review and to meet the standards expected by the scientific community, especially for a publication in Nature Communications, it is essential that the authors provide both the QC data (minimum) as supplementary data and the processed individual data (peptide and protein level) of this study for review purposes only (since it is restricted by ethical constraints).

These datasets should also be made available, so data tables are reachable with reviewer login credentials. I propose a similar strategy where the raw data and data tables are published with the same comment used in the Messner 2020 study: "The raw data of the acquired commercial plasma and serum control samples within the GS study have been deposited to the ProteomeXchange Consortium via PRIDE". Data tables for all individuals should thus be available for the review process only, but the QC raw data collected throughout the study is enough to assess the overall analytical quality of this study.

Aside from this key aspect, which makes the review process conceptual, it offers valuable insights into the non-genetic factors, notably lifestyle and medication, affecting the plasma proteome. A key finding is the influence of hormonal contraceptives on the plasma proteome, surpassing other medications or covariates, and challenging some existing notions about the long-term effects of hormonal therapies. This underscores the necessity of considering hormonal contraceptive use in epidemiological or clinical studies to prevent misinterpretations related to sex or age.

The study would benefit from additional validation of the findings, through orthogonal methods or the use of spiked standards. This is crucial to confirm the accuracy and precision of the findings, particularly for significant variables like the LPA protein with its kringle domain that introduces a genetic effect, based on what peptides are used to quantify the protein. Without access to the raw data, it's challenging to verify these findings, underscoring the necessity for data sharing.

The Git repository appears comprehensive, but a thorough assessment is limited without the corresponding data. The code, in isolation, does only provide a snapshot of the study's methodology without any input data.

In conclusion, the study is conceptually strong and timely, with potential implications for clinical proteomics. However, to proceed with publication and proper review from my side of their work, it is essential that the authors address these concerns, particularly regarding data transparency and methodological validation. These steps are crucial for a thorough evaluation of the study's quality and to uphold the standards of scientific research and publication. Please note that I fully respect individual samples and individuals' right to remain de-identified. I thus believe that this does not apply to QC samples pooled samples that have been quantified in this study.

Reviewer #2

(Remarks to the Author)

The article "Pervasive Influence of Hormonal Contraceptives on the Human Plasma Proteome in a Broad Population Study"

has novel findings about the outstanding effect of hormonal contraceptives (HC) on human plasma proteome. The impact of these drugs is so exceptional that it distinguishes HC users from the rest of the population and may interfere with the results of epidemiological studies. The study population of 3632 subjects is impressive, and proteomic analysis is a novel way to investigate broadly the effects of different medications.

For me, as a clinician and researcher, the most interesting findings of the study were these:

- The effect of the HC was found to be so outstanding that it separated the users from the rest of the population. This is significant as HCs are mainly used just for contraception, and they would not need to have any other side effects.
- The effect of HC should be considered as a confounding factor in epidemiological studies with large cohorts.
- The proteome of previous users of the HC did not differ from the women who had never used HC. Therefore, it seems that the effect could disappear when the HC use is stopped.
- Hormonal replacement therapy (HRT) did not majorly affect plasma proteome. The significant difference between HC and HRT is the type of estrogen used in these preparations; HRT has natural estrogen, whereas most HC preparations contain very potent ethinyl estradiol (EE). This leaves the question about newer HCs with natural estrogens: are they neutral, too?

The findings of this article are significant and interesting in the field of epidemiology and have major clinical importance. However, the audience most likely misses these important points in the article's current form.

Think about your audience: who are you writing for? Scientists familiar with proteomic analyses can most likely read and understand this paper even though some modifications could help them. Most epidemiologists and clinicians are not familiar with proteomics at all. If you want them to get interested, read, and understand your work, it should be written as clearly as possible.

Here are some recommendations for revising the manuscript.

1. What is your main message to the readers? Keep it clear when writing the whole article and address it as a conclusion at the end of the discussion.
2. I would prefer the more traditional order of the chapters: Introduction, materials & methods, results, and discussion. Put all complicated methodological explanations to mat&met and leave them out from the results chapter. Readers unfamiliar with proteomics will skip mat&met and read the results and discussion. Present your findings there as clearly as possible and tell what you think is the significance of those results.
3. Could you see more precisely what kind of hormonal contraceptives were in use? EE combinations, natural estrogen (estradiol, estradiol valerate, or estetrol) combinations, or progestin-only pills? If not, please also tell that in your article. It might have a significant difference, so at least discuss that in the discussion. It's been shown previously that the estradiol valerate combination and progestin-only pill were more neutral than the EE combination regarding the effects on plasma proteome (your reference 52).
4. Vaginal combined contraceptives are shown to have similar effects on metabolism and inflammations as oral contraceptives and hormonal patches (reference Piltonen et al. 2012: 10.1093/HUMREP/DES225). Therefore, it was not a surprise they showed a similar profile with oral HC, and it could have even been assumed. Please consider this when revising your text.
5. Avoid using gene abbreviations in your text. If you need to present exact proteins in your text, I recommend using the names of the proteins, as they might tell something to the readers. Plain gene abbreviations won't mean anything to clinicians. Include protein names in your tables, too.
6. Table 1: Do you have information about smoking? Please add numerical values for BMI limits as you do in the text, for example, in line 259.
7. Line 144: GO:BP pathway, can you open the abbreviation?
8. Figures: Please summarize the message of the figure in your figure text. Readers unfamiliar with proteomics won't get the point of your figures otherwise.
9. Figure 3: What is the key message of figures A & B? What do they tell? Could they be left out? C seems more clinically relevant, but could you explain more clearly what it tells? What do 1-2, 3-2 & 4-3 mean? Where can you see BMI categories?
10. Figure 4: BMI 4 vs 2; please explain these numbers in the figure text.
11. Line 207: "As significance criteria we required for categorical variables, in addition to a statistical significance level, also the observed average difference in abundances to be larger than the CV of the respective protein determined on study specific quality control samples measured in the same data set." A very long sentence that is difficult to understand.
12. Discussion: The first paragraph should draw together the study's most important findings; please modify your text accordingly.
13. Think about your paragraphs in the discussion. Modify your text so that you write about one entity in one paragraph. For example, the paragraph beginning from line 382 contains many different things, making it a bit confusing.
14. You could add one independent paragraph for strengths and weaknesses, making the text more coherent.
15. Line 359 onwards: "Using an HPLC setup with two binary pump systems, one gradient and one wash/equilibration pump, reduced overheads to 1.8 min and eventually allowed one operator to run 3 LC-MS batches (3x 96-well plates each) per week and to complete the measurements within 4 weeks of instrument time." repeating of methods, remove from the discussion? Or do you think this is a strength of your work? If you do, please be more clear about why.
16. Chapter beginning in line 382: Numbers 12/22, 63/91, and 28/29 do not tell much. Could you maybe tell it shorter, like "about half of the sex-associated proteins identified in our study have been described using other methods" or "two-thirds of age-associated protein abundances in our dataset have been previously associated with aging in the literature"? Again, gene abbreviations tell something only for rare readers and almost nothing for clinicians. Could you tell about the findings in some other way?
17. I hope that in the final manuscript, you will talk about the HC effect on proteome earlier in the discussion as it is the study's main finding (maybe in the second paragraph if the first just summarizes the results). Please be clear where you talk

about hormonal contraceptives, hormonal replacement therapy, and your results vs other studies. Right now, it is not clear enough.

18. In conclusion, please do not repeat what you did. Clarify here the main thing you want to say with your article. What should the reader remember after reading this paper?

19. Materials and methods, Study cohort section: In this article, you talk about proteomics: why do you prescribe other measurements like antithrombin and lipids in detail? Please delete the information that is not handled in this article.

20. Supplementary dataset Excel files: do you have a different file where you tell what all the columns mean? That kind of file is needed so that others can use this data.

21. Would you have some familiar clinician (maybe a gynecologist or someone doing epidemiological research) not included in this work who could give you feedback about the revised article? If they can tell what the study's main message was after reading the manuscript, it gives you a good sign of clarity.

Version 1:

Reviewer comments:

Reviewer #1

(Remarks to the Author)

In "Pervasive Influence of Hormonal Contraceptives on the Human Plasma Proteome in a Broad Population Study," Dordevic et al. describe the use of an MS-based analysis of circulating proteins in population cohorts, where they identify contraceptives to influence the proteomes. The study brings forward an important aspect and is a useful reminder to consider medication use in other blood proteomics studies.

Coming in as a reviewer after the first round of revision, the authors have, from my perspective, already addressed a few comments that likely improved the manuscript. It is fine and common for data not to be shared openly (thanks to GDPR). In summary, the quality of the study and results warrant that this work should - undoubtedly - be published in Communications Medicine. However, the manuscript is still too immature to be accepted in its current form.

Here are my major concerns and suggestions:

The authors have conducted a global analysis and observed a subgroup of donors to deviate from the remaining population. This discovery defines the direction of further investigation. Hence, it must be grounded in objective/statistical criteria. However, there is no explanation of which statistical approach was used to define membership of the subgroup. This is an essential step that needs to be defined and justified. Does this subgroup remain in other clustering approaches and when looking only at women?

It remains unclear if the authors intended to search for medication effects before executing the analysis or if their main objective was to explore the proteome in an unbiased manner (and then follow any possible leads). It must be clear what the underlying hypothesis was when they started this investigation to avoid the character of a fishing expedition. This should become clear from the abstract and introduction.

Finding contraceptives as proteome modulators also shifts the focus away from males and women > 40. Any analysis on contraceptives must only be conducted in women, thus, discussing the analysis of the whole cohort(s) adds unnecessary confusion.

Fig 1A is an essential selling point for the study but is too overloaded to be meaningful. Split the PCA into two: one for samples and one for protein loadings. Label sample PCA by sex. Create PCA figures per sex to demonstrate if the subgroup remains. What's the message and value of the protein PCA?

Fig 1B: Use sample PC1 values as a continuous variable and plot it against age. The current boxplot categorization is not meaningful. Prepare one figure for females and males (see Fig 1A).

Fig 2: Please rotate all numbers to horizontal orientations and explain why the BMI categories have been used. Please lift the sex-specific proteins being affected by contraceptives. Ideally, present the overlap in associations of the traits. Also, test if the age of the samples had an influence on the data.

Fig 4 is useless in its current form for me. Is the ranked order based on author preference or something else? What message should the reader take home from this? I think you should remove this figure.

Fig 5 is very good. Please rotate all numbers to horizontal orientations. It is safe to remove the header from B and E.

Fig 6 is fine, but the font size was irritating. I prefer to spell out BE and EE as these are important parameters. What fit was used to calculate the slope? Is the difference in slope statistically significant?

The analysis of current and previous contraceptive users must include age and BMI. This ensures that these two influential traits do not compromise the study (e.g., younger women are likely to be leaner and current users than older women). Also, consider the type of contraceptive as in Fig 6. Please remove women who donated blood after giving birth and if they did not

restart the use of contraceptives.

Please discuss the results of a recent preprint (<https://doi.org/10.1101/2024.08.12.607634>) in which TFF3 was highlighted (and replicated from earlier studies).

Are there any findings (proteins) related to the female reproductive system (as compared to the study above), or is the highly abundant proteome not built on proteins originating from female organs?

Other concerns:

Many sections in the abstract and introduction don't align with the study's main objective and message. These appear as artifacts of previous versions of the manuscript. Overall, the story would be much easier to read and digest if unrelated aspects (lines 65-96) were replaced by more topic-specific information. Please remove any unnecessary and redundant technical side stories to mature the manuscript further. For example, focus on the main medical message rather than the material or method.

A supplementary with > 100 figures seems unnecessary, and it was highly inconvenient to find the added figures that would be of value. Consider providing separate files for ease of access. Consider creating a website (e.g., Shiny app) where users can interact with the data.

All figure legends are complicated and lack a short descriptive title.

The demographics of the second cohort were missing. Please add and focus on females only.

Present global CVs, correlations, and such in density plots to provide a better overview of the data. Use IQR instead of CVs when talking about the variance in a population (= across samples), since CVs are meant to convey precision.

Avoid the term "signal" and use "data" instead (also in supplementary).

The second cohort is built on serum samples, so the title should read "Human Circulating Proteome" instead.

Line 548 should read: "...one of the larger..."

Pasting tables into the text flow did not help; add to the end.

Reviewer #2

(Remarks to the Author)

After the first round of review, the authors have done major work on the manuscript. The original manuscript has improved greatly, and the authors have really taken into account the reviewers' comments. The work has also improved in readability, and it is now easier for the wider audience to read. This is great work, and now the results and discussion are more clear. Furthermore, adding the subanalysis of contraceptives with differing estrogens adds scientific value to the work. Even though I still might have formulated the discussion a bit differently, the work has improved greatly, and I think that the clinical/epidemiological value of the results is now clear enough to publish.

Version 2:

Reviewer comments:

Reviewer #1

(Remarks to the Author)

The authors have done excellent work addressing all my concerns. I only have the following minor suggestions, which I think should not require another round of review and can be handled by the editorial team. Still, there were some minor artifacts from earlier versions, requiring some final fine-tuning.

1) Title: The element "Broad Population Study" is not a good fit. Two independent cohorts were used in the end. Suggest to revise in the direction of "population-based studies reveal..."

2) Abstract: As indicated above, both cohorts should be mentioned (it's a strength). The current "Background" description of CHRIS should be part of the "Methods," and the new "Background" should describe the more general use of population cohorts to explore influential, modifiable, and non-modifiable factors in the circulating proteome that assist us to describe human health.

3) Discussion: (i) Please add the limitation of only using European-based cohorts, hence missing to determine how the discovered effects would influence the circulating proteome in other populations and ancestries. (ii) Please discuss the mechanism of OC-induced proteome changes, for example, via the liver's CYP450 system. (iv) Please dare to speculate

how drug-induced liver activating could drive their observations. (v) Please refer to this previous metabolism-focussed OC study <https://doi.org/10.1093/ije/dyw147>, in which similar observations were made.

Overall, this is an excellent study that presents important insights into the impact of regular and common medication use in females.

Jochen Schwenk

Berlin, Bolzano, 08.07.2024

Dear Reviewers,

We thank the reviewers for their thorough review and the helpful suggestions. By addressing the points raised, we believe the original manuscript to be greatly improved.

Before answering the individual points, we provide some general remarks and descriptions of the major changes in comparison to the first submission:

Reviewer 1 asked about data availability. All data generated in this study is made available to the community. However, due to the legal situation in Europe, and more specifically the data privacy laws in Italy, the sharing of any individual level human data requires a General Data Protection Regulation (GDPR) compliant legal framework, i.e. a specific data sharing agreement between legal entities (the sender and the receiver) is mandatory. This is a common situation intended to prevent misuse of individual human data. For example, the study's access committee needs to make sure that the data is not used for other purposes than the ones covered by the participants' consent forms. This situation applies not only to us but to epidemiological studies in general. In other words, data is made public, and any reader can get access to the data, but because individual human data is involved, accessing the data requires following a procedure.

We evaluated, together with our legal and ethics departments and the institute's data protection officer, whether exceptions can be made, but unfortunately, this was not deemed possible for the individual human data. We were however allowed to deposit, without access restrictions, the QC samples from the study to a public repository (PRIDE), which allows us to validate all technical aspects. Furthermore, we have added a paragraph explaining the procedure for accessing the data. A step-by-step guide is provided online (<https://chrisportal.eurac.edu/how-to-apply>).

Another comment related to validation of our findings. Here, a major improvement of our study compared to its first version, is that we have included an analysis of proteomics data generated for an independent cohort. More specifically, we validated the signal stemming from hormonal contraceptives in serum proteome samples of the Berlin Ageing study II (BASE-II) [<https://doi.org/10.1101/2024.06.22.24309293>]. The results obtained validate and solidify the impact of hormonal contraceptives on the plasma proteome. In addition, we also expanded the comparison of our results with previous publications addressing individual proteins, with orthogonal - in most cases classic - protein quantification techniques. Finally, following the suggestion from reviewer 2, we restructured the manuscript and included an analysis in which we evaluate the influence of different combined oral contraceptives (more precisely, different

types of estrogens), on the plasma proteome. We believe these additions improve the quality and relevance of our work considerably.

We hope you find our substantially revised manuscript now suitable for publication.

For all authors,

Johannes Rainer and Markus Ralser

Below we address the individual points raised (shown with blue, italics font) with our replies.

Reviewer #1 (Remarks to the Author):

This study by Dordevic et al presents a useful approach to quantitative plasma proteomics, quantifying 148 proteins in a cohort of 3 632 participants using ultra-high-throughput mass spectrometry with SWATH readout. The research aligns well with the previous study introducing the concept of using SWATH with extremely short gradients, Messner et al. (2020). This has the potential to move rapid plasma profiling in clinical settings and the authors show that this technology makes it feasible to run thousands of plasma samples from a clinical cohort.

We thank the reviewer for their overall assessment of the manuscript

However, while the concept and methodology are promising, there are critical areas where the study must be strengthened. The lack of data availability is a significant concern. For a robust peer review and to meet the standards expected by the scientific community, especially for a publication in Nature Communications , it is essential that the authors provide both the QC data (minimum) as supplementary data and the processed individual data (peptide and protein level) of this study for review purposes only (since it is restricted by ethical constraints).

These datasets should also be made available, so data tables are reachable with reviewer login credentials. I propose a similar strategy where the raw data and data tables are published with the same comment used in the Messner 2020 study: "The raw data of the acquired commercial plasma and serum control samples within the GS study have been deposited to the

ProteomeXchange Consortium via PRIDE". Data tables for all individuals should thus be available for the review process only, but the QC raw data collected throughout the study is enough to assess the overall analytical quality of this study.

The Git repository appears comprehensive, but a thorough assessment is limited without the corresponding data. The code, in isolation, does only provide a snapshot of the study's methodology without any input data.

We thank the reviewer for this comment. As aforementioned, we are very much aligned with the reviewer about data access and transparency, and indeed all data produced in this study is made publicly available and can be re-used by the community, e.g. in follow-up studies, and for reproducing our results. No data is withheld from the community. As aforementioned, like in other epidemiological studies, however, personal and individual human data is involved, and due to the applicable Italian law, it is not permitted for us to put primary human data for anonymous or unrestricted download, i.e.. without a binding legal agreement in place. These agreements exist to prevent misuse of the human data, for example, to use the data for other purposes than the ones the study participants have consented to, or, for a use that has not been approved by a recognized ethics committee. As it is common practice also on other epidemiological cohorts (e.g. UK Biobank), accessing the data thus requires the set-up of a data sharing agreement under the General Data Protection Regulation (GDPR) compliant legal framework. We explain the procedure in the revised document, and online at <https://chrisportal.eurac.edu/downloads> with the access policy described in detail in this document: https://webassets.eurac.edu/31538/1674124649-data-and-sample-access-policy_v2.pdf

Having said that, we make everything possible that any reader can, even without such an agreement in place, evaluate and reproduce the technical aspects of our study. We provide considerable information on data quality, which includes the distribution of signals for each quantified protein as well as the coefficients of variation for each protein from study-specific, repeatedly measured, QC samples. As suggested by the reviewer, we now also deposited the data for the study's QC samples in the public repository PRIDE with accession number PXD052861 and PXD052892, including raw measurement data, the spectral library used for annotation of MS-precursors and the corresponding software output.

Aside from this key aspect, which makes the review process conceptual, it offers valuable insights into the non-genetic factors, notably lifestyle and medication, affecting the plasma proteome. A key finding is the influence of hormonal contraceptives on the plasma proteome, surpassing other medications or covariates, and challenging some existing notions about the long-term effects of hormonal therapies. This underscores the necessity of considering hormonal contraceptive use in epidemiological or clinical studies to prevent misinterpretations related to sex or age.

The study would benefit from additional validation of the findings, through orthogonal methods or the use of spiked standards. This is crucial to confirm the accuracy and precision of the findings, particularly for significant variables like the LPA protein with its kringle domain that introduces a genetic effect, based on what peptides are used to quantify the protein. Without access to the raw data, it's challenging to verify these findings, underscoring the necessity for data sharing.

We agreed with the reviewers that additional validation of our manuscript would increase its value. Therefore, we conducted an additional analysis evaluating the impact of hormonal contraceptives on serum proteomics data from an additional study, the Berlin aging study II (BASE II) study. The results obtained in this analysis did validate our results with data generated with a different MS instrument on samples from an independent cohort. The results are presented in the results section “Hormonal Contraceptive Use Induces Similar Proteomic Changes in an Independent Cohort”, and in Figure 5, as well as discussed in the discussion section. In our opinion this independent replication consolidates the strong effect of hormonal contraceptives seen in our data.

Moreover, we also expanded the comparison of our results with previous literature results, in which proteins were quantified using different, targeted, methods. These are provided as supplementary table S49 and presented in the discussion section.

In conclusion, the study is conceptually strong and timely, with potential implications for clinical proteomics. However, to proceed with publication and proper review from my side of their work, it is essential that the authors address these concerns, particularly regarding data transparency and methodological validation. These steps are crucial for a thorough evaluation of the study's quality and to uphold the standards of scientific research and publication. Please note that I fully respect individual samples and individuals' right to remain de-identified. I thus believe that this does not apply to QC samples pooled samples that have been quantified in this study.

Reviewer #2 (Remarks to the Author):

The article "Pervasive Influence of Hormonal Contraceptives on the Human Plasma Proteome in a Broad Population Study" has novel findings about the outstanding effect of hormonal contraceptives (HC) on human plasma proteome. The impact of these drugs is so exceptional that it distinguishes HC users from the rest of the population and may interfere with the results of epidemiological studies. The study population of 3632 subjects is impressive, and proteomic analysis is a novel way to investigate broadly the effects of different medications.

For me, as a clinician and researcher, the most interesting findings of the study were these:

- The effect of the HC was found to be so outstanding that it separated the users from the rest of the population. This is significant as HCs are mainly used just for contraception, and they would not need to have any other side effects.*
- The effect of HC should be considered as a confounding factor in epidemiological studies with large cohorts.*
- The proteome of previous users of the HC did not differ from the women who had never used HC. Therefore, it seems that the effect could disappear when the HC use is stopped.*
- Hormonal replacement therapy (HRT) did not majorly affect plasma proteome. The significant difference between HC and HRT is the type of estrogen used in these preparations; HRT has natural estrogen, whereas most HC preparations contain very potent ethinyl estradiol (EE). This leaves the question about newer HCs with natural estrogens: are they neutral, too?*

The findings of this article are significant and interesting in the field of epidemiology and have major clinical importance.

We thank the Reviewer for their work and their positive assessment of our study.

However, the audience most likely misses these important points in the article's current form. Think about your audience: who are you writing for? Scientists familiar with proteomic analyses can most likely read and understand this paper even though some modifications could help them. Most epidemiologists and clinicians are not familiar with proteomics at all. If you want them to get interested, read, and understand your work, it should be written as clearly as possible.

Here are some recommendations for revising the manuscript.

- 1. What is your main message to the readers? Keep it clear when writing the whole article and address it as a conclusion at the end of the discussion.*

We thank the reviewer for their productive comments, to help our manuscript to become more accessible. We have taken these onboard, and have significantly reworked the manuscript text in order to make it maximally accessible for the readers.

2. I would prefer the more traditional order of the chapters: Introduction, materials & methods, results, and discussion. Put all complicated methodological explanations to mat&met and leave them out from the results chapter. Readers unfamiliar with proteomics will skip mat&met and read the results and discussion. Present your findings there as clearly as possible and tell what you think is the significance of those results.

We restructured the manuscript in the suggested order of sections. We removed many technical details from the results and discussion sections, where appropriate. However, we left some information on the proteomics data set generation in the results section as in part the novelty and impact of our manuscript stems not only from the specific findings but also from the methodology.

3. Could you see more precisely what kind of hormonal contraceptives were in use? EE combinations, natural estrogen (estradiol, estradiol valerate, or estetrol) combinations, or progestin-only pills? If not, please also tell that in your article. It might have a significant difference, so at least discuss that in the discussion. It's been shown previously that the estradiol valerate combination and progestin-only pill were more neutral than the EE combination regarding the effects on plasma proteome (your reference 52).

We thank the reviewer for this suggestion. We re-evaluated the available medication information, categorized oral hormonal contraceptives based on the included estrogen into combined oral combinations (COC) with ethinylestradiol (n = 248), bioidentical estrogen (n = 17) and progestogen only preparations (n = 6) and performed an additional analysis comparing the effect of these different COCs on the plasma proteome. As indicated by the reviewer and shown in the mentioned (and now added) reference, we found also in our data combinations containing bioidentical estrogen to have an about 3-times weaker effect on the plasma proteome than combinations containing ethinylestradiol (new Figure 6, Supplementary Table S48, Supplementary Figure S157). We added this additional analysis to the manuscript.

4. Vaginal combined contraceptives are shown to have similar effects on metabolism and inflammations as oral contraceptives and hormonal patches (reference Piltonen et al. 2012:

10.1093/HUMREP/DES225). Therefore, it was not a surprise they showed a similar profile with oral HC, and it could have even been assumed. Please consider this when revising your text.

We thank the review for this input. We added the suggested reference and reformulated the respective sentence in the discussion section.

5. Avoid using gene abbreviations in your text. If you need to present exact proteins in your text, I recommend using the names of the proteins, as they might tell something to the readers. Plain gene abbreviations won't mean anything to clinicians. Include protein names in your tables, too.

We replaced gene symbols by protein names if proteins are discussed in detail (e.g. Angiotensinogen; AGT in the results section) or added plain proteins in addition to the official HGNC gene symbols throughout the text (according to the Communications Medicine guidelines, HGNC symbols should be reported for proteins/genes). In addition, we added protein names to all tables in the Manuscript and Supplement.

6. Table 1: Do you have information about smoking? Please add numerical values for BMI limits as you do in the text, for example, in line 259.

Numerical values for BMI have been added to Table 1, we could however not address smoking status as part of the present study.

7. Line 144: GO:BP pathway, can you open the abbreviation?

The abbreviation refers to the Gene Ontology Terms for Biological Processes. Explanations have been added to the text.

8. Figures: Please summarize the message of the figure in your figure text. Readers unfamiliar with proteomics won't get the point of your figures otherwise.

Thanks for this input. We have extended the figure legends (also based on examples from other publications from Communication Medicine) to better describe the content and information shown in the plots.

9. Figure 3: What is the key message of figures A & B? What do they tell? Could they be left out? C seems more clinically relevant, but could you explain more clearly what it tells? What do 1-2, 3-2 & 4-3 mean? Where can you see BMI categories?

The figure (now Figure 2 after restructuring of the manuscript) shows 2 Volcano plots representing differential protein expression depending on sex and age. Each point is one protein with the strength of association (coefficient) shown on the x-axis and its significance on the y-axis. We use these plots to visualize proteomic changes in large scale studies; they provide an overview of the full results in a single plot. Labels 1-2, 3-2 and 4-2 refer to the comparison of the clinically relevant BMI categories (underweight(1), overweight(3) and obese (4)) vs the normal/healthy BMI category (2). We expanded the figure legends and provided more information/description of the plots.

10. Figure 4: BMI 4 vs 2; please explain these numbers in the figure text.

We apologize for this omission. Numbers are now explained in the figure legend. See also answer to question 9.

11. Line 207: "As significance criteria we required for categorical variables, in addition to a statistical significance level, also the observed average difference in abundances to be larger than the CV of the respective protein determined on study specific quality control samples measured in the same data set." A very long sentence that is difficult to understand.

This sentence was reformulated and split into two separate sentences.

12. Discussion: The first paragraph should draw together the study's most important findings; please modify your text accordingly.

We restructured and re-ordered the results of the manuscript (also related to point 21) to better follow the analysis workflow: starting from the largest possible data set we provide general information on detected proteins and largest variance explained. Then we investigate associations with general epidemiological traits to allow a first validation of the data based on comparison with previous literature. This analysis also includes the sensitivity analysis emphasizing the need to account for HCU in plasma proteomics studies. We then investigate the influence of medication on the same, large data set which shows that hormonal contraceptives have by far the largest influence. We conclude the analysis focusing on this medication, performing additional analyses on specific data subsets. This includes also the new comparison

of the effect from different combined oral contraceptives. We structured the discussion now in the same order as the results and believe this order will make it easier for the common reader.

13. Think about your paragraphs in the discussion. Modify your text so that you write about one entity in one paragraph. For example, the paragraph beginning from line 382 contains many different things, making it a bit confusing.

We apologize that our discussion section was indeed not structured well. We re-evaluated it to ensure each paragraph being within itself consistent and have also revised the specific paragraph the Reviewer addressed.

14. You could add one independent paragraph for strengths and weaknesses, making the text more coherent.

We split the information on the limitation of the present data into a separate paragraph, discussing, as suggested by the reviewer, strengths and weaknesses of the present data.

15. Line 359 onwards: "Using an HPLC setup with two binary pump systems, one gradient and one wash/equilibration pump, reduced overheads to 1.8 min and eventually allowed one operator to run 3 LC-MS batches (3x 96-well plates each) per week and to complete the measurements within 4 weeks of instrument time." repeating of methods, remove from the discussion? Or do you think this is a strength of your work? If you do, please be more clear about why.

We reformulated the sentence. As aforementioned, part of the value of the manuscript is also its technical aspects, as the methodology used, allowing us to scale proteomic measurements, is not common. Conventional mass spectrometry-based proteomics approaches were of much lower throughput, and more prone to batch effects and stochastic samples, limiting the application of these techniques on large sample series. Our particular settings/measurement setup guaranteed generation of a low variance, high quality data set at acceptable costs.

16. Chapter beginning in line 382: Numbers 12/22, 63/91, and 28/29 do not tell much. Could you maybe tell it shorter, like "about half of the sex-associated proteins identified in our study have been described using other methods" or "two-thirds of age-associated protein abundances in our dataset have been previously associated with aging in the literature"? Again, gene abbreviations tell something only for rare readers and almost nothing for clinicians. Could you tell about the findings in some other way?

We reformulated that paragraph in the discussion as suggested by the reviewer.

17. I hope that in the final manuscript, you will talk about the HC effect on proteome earlier in the discussion as it is the study's main finding (maybe in the second paragraph if the first just summarizes the results). Please be clear where you talk about hormonal contraceptives, hormonal replacement therapy, and your results vs other studies. Right now, it is not clear enough.

We appreciate the feedback and considered this in the reformulation of paragraphs in the discussion. We take great efforts in clearly distinguishing hormonal contraceptives, hormonal replacement therapy, and our results vs other studies.

18. In conclusion, please do not repeat what you did. Clarify here the main thing you want to say with your article. What should the reader remember after reading this paper?

We reformulated the last paragraph in the discussion section accordingly.

19. Materials and methods, Study cohort section: In this article, you talk about proteomics: why do you prescribe other measurements like antithrombin and lipids in detail? Please delete the information that is not handled in this article.

These data were used in the comparison of MS-based proteomics results with these accredited measurements (in the results section and supplementary material). The results give confidence in our proteomic data, but also show that proteomics is complementary to the established assays.

20. Supplementary dataset Excel files: do you have a different file where you tell what all the columns mean? That kind of file is needed so that others can use this data.

We appreciate this suggestion and have now added a second table (i.e., sheet within each Excel file) with a description of the individual columns.

21. Would you have some familiar clinician (maybe a gynecologist or someone doing epidemiological research) not included in this work who could give you feedback about the revised article? If they can tell what the study's main message was after reading the manuscript, it gives you a good sign of clarity.

We let a gynecologist read the revised article and she provided valuable feedback. According to her feedback, she will indeed consider the results, in particular on the different types of combined hormonal contraceptives, in her clinical routine.

Dear Reviewers,

We thank the reviewers for the detailed review of our manuscript. We addressed all raised points and comments and hope you find our revised manuscript suitable for publication. All changes have been highlighted in yellow in the revised manuscript. In the individual replies below, we also provide the line numbers of the respective changes. These are based on the Word document we submitted but might have been changed during the automatic manuscript conversion process. We apologize if the line numbers might differ.

For all authors,

Johannes Rainer and Markus Ralser

Below we address the individual points of the reviewers (italics font) with our replies (shown with blue font).

Reviewers' comments:

Reviewer #1 (Remarks to the Author):

In "Pervasive Influence of Hormonal Contraceptives on the Human Plasma Proteome in a Broad Population Study," Dordevic et al. describe the use of an MS-based analysis of circulating proteins in population cohorts, where they identify contraceptives to influence the proteomes. The study brings forward an important aspect and is a useful reminder to consider medication use in other blood proteomics studies.

Coming in as a reviewer after the first round of revision, the authors have, from my perspective, already addressed a few comments that likely improved the manuscript. It is fine and common for data not to be shared openly (thanks to GDPR). In summary, the quality of the study and results warrant that this work should - undoubtedly - be published in Communications Medicine. However, the manuscript is still too immature to be accepted in its current form.

Here are my major concerns and suggestions:

The authors have conducted a global analysis and observed a subgroup of donors to deviate from the remaining population. This discovery defines the direction of further investigation. Hence, it must be grounded in objective/statistical criteria. However, there is no explanation of which statistical approach was used to define membership of the subgroup. This is an essential step that needs to be defined and justified. Does this subgroup remain in other clustering approaches and when looking only at women?

We thank the reviewer for raising this point. Indeed, these analyses and results are important, and we apologize that they were missing in the original manuscript. We have now conducted the missing formal analysis, based on multiple linear regression, to systematically investigate variables that are related to principal component 1. The analysis confirmed hormonal contraceptive related medications to be the strongest associated factors, followed by age, sex and BMI category 4 (obesity). We updated the results part (lines 313-324, last paragraph of the “Protein Coverage and Variation in the CHRIS Cohort” results section) and added the respective result table to the supplement (Supplementary Table S5). In addition, we performed the requested separate analyses on the male and female study participants and confirmed the same separation in the PCA on the plasma proteome from women, while it is not present in the PCA on men (Supplementary Figures S3 and S4).

It remains unclear if the authors intended to search for medication effects before executing the analysis or if their main objective was to explore the proteome in an unbiased manner (and then follow any possible leads). It must be clear what the underlying hypothesis was when they started this investigation to avoid the character of a fishing expedition. This should become clear from the abstract and introduction.

We started this analysis without being biased about a particular outcome; the finding that the use of hormonal contraceptives has such a strong and dominating impact on the plasma proteome came as a surprise. A broad literature research on plasma proteomic studies did then reveal that this situation is not commonly known. When comparing the response to different common medications we found hormonal contraceptives to have the largest effect on the proteome. If not accounted for, this effect can often be misconstrued as age and sex specific effects. As requested, we amended the abstract and introduction of the revised manuscript to clearly describe the hypothesis and analysis approach.

Finding contraceptives as proteome modulators also shifts the focus away from males and women > 40. Any analysis on contraceptives must only be conducted in women, thus, discussing the analysis of the whole cohort(s) adds unnecessary confusion.

We agree with the reviewer that identification of proteins associated with hormonal contraceptives needs to be performed in the data subset of women of age < 40. The result section for this particular young-female-only analysis (results sections “Oral Hormonal Contraceptives Shape the Plasma Proteome in Female Study Participants” and “Hormonal Contraceptive Use Induces Similar Proteomic Changes in an Independent Cohort”), as well as all other analyses related to contraceptives (results sections “Combined Hormonal Contraceptives Containing Ethinylestradiol Have a Stronger Effect Than Those with Natural Estrogens” and “No Long-Lasting Effects of Hormonal Contraceptives on the

Plasma Proteome Observed”) already clearly describe the data subset on which the analysis was performed. To further clarify we now also updated the respective paragraph in the discussion section (lines 563-565, paragraph number 4 in the Discussion section).

Fig 1A is an essential selling point for the study but is too overloaded to be meaningful. Split the PCA into two: one for samples and one for protein loadings. Label sample PCA by sex. Create PCA figures per sex to demonstrate if the subgroup remains. What's the message and value of the protein PCA?

We split the PCA in Fig 1 as suggested into two panels, one showing the sample grouping (new Fig 1A) and one with the PCA loadings of the proteins (new Fig 1B). We also performed the PCA separately for men and women and added these to the supplement (Supplementary Figures S3 and S4) as well as for the validation in BASE-II (Supplementary Figure S10). Indeed, as mentioned above, the same separation is visible only in the PCA on female participants. We refer now also to the protein PCA in the results part (lines 313-314, last paragraph in the “Protein Coverage and Variation in the CHRIS Cohort” results section). The PCA loadings plot shows which proteins are driving the observed separation, which are then later in the result section “Oral Hormonal Contraceptives Shape the Plasma Proteome in Female Study Participants” identified as being HCU associated.

Fig 1B: Use sample PC1 values as a continuous variable and plot it against age. The current boxplot categorization is not meaningful. Prepare one figure for females and males (see Fig 1A).

As suggested by the reviewer, we replaced the boxplots with a new scatter plot showing the relationship between PC1 and age (new Fig 1C). This plot and the low p-value of the linear model fitted to PC1 and age (Supplementary Table S5) clearly shows the age-relationship of PC1. Figures for females and males separately are shown in the supplement (Supplementary Figures S3 and S4, right panels).

Fig 2: Please rotate all numbers to horizontal orientations and explain why the BMI categories have been used. Please lift the sex-specific proteins being affected by contraceptives. Ideally, present the overlap in associations of the traits. Also, test if the age of the samples had an influence on the data.

We rotated all numbers in all figures to horizontal orientations. We used BMI categories as defined by the WHO ([https://www.who.int/data/nutrition/nlis/info/malnutrition-in-women#:~:text=BMI%20is%20a%20simple%20index,metres%20\(kg%2Fm2\)](https://www.who.int/data/nutrition/nlis/info/malnutrition-in-women#:~:text=BMI%20is%20a%20simple%20index,metres%20(kg%2Fm2))) since they are intuitive for both healthcare professionals and the general public. We included, as requested, an additional plot (new Figure 2D) showing the overlap in associations of the traits. We did not highlight the sex-associated proteins being affected by HCU as the volcano became too busy. This information is available in the Supplement (Supplementary Tables S12 and S13). Regarding the age of the samples, we performed additional analyses but did not find any influence on the data: no correlation was seen between sample age and PC1 (we added this information to the respective part in the results section (lines 323-324, last paragraph in the “Protein Coverage and Variation in the CHRIS Cohort” results section); Spearman’s rho = -0.03) and an additional sensitivity analysis with and without sample age

as an additional covariate showed that it did not have any influence on the main findings. The results from this sensitivity analysis are shown below: the coefficients from the analysis with (y-axis) and without sample age as covariate (x-axis) are almost identical.

Fig 4 is useless in its current form for me. Is the ranked order based on author preference or something else? What message should the reader take home from this? I think you should remove this figure.

This figure complements and completes Table 2 which lists the number of significant proteins per medication. This figure illustrates the proteins that are affected by a medication and to what extent. In addition, it allows the reader to identify medications that share similar protein signatures. The column and row grouping reflects a hierarchical cluster analysis (Ward-based clustering of an Euclidean dissimilarity matrix). That said, we agree with the reviewer that including this figure in the main manuscript may not be necessary. To address the reviewer's point, we have removed the figure from

the main text and moved it to the Supplement as the new Supplementary Figure S7. Also, we expanded the figure legend for this figure to include a description of the row and column ordering.

Fig 5 is very good. Please rotate all numbers to horizontal orientations. It is safe to remove the header from B and E.

We rotated all numbers to horizontal orientations (in all Figures) and removed the headers from B and E from Figure 5 (Figure 4 in the revised manuscript).

Fig 6 is fine, but the font size was irritating. I prefer to spell out BE and EE as these are important parameters. What fit was used to calculate the slope? Is the difference in slope statistically significant?

We updated this figure (now Figure 5 in the revised manuscript), fixed the font sizes and spelled out ethinylestradiol and bioidentical estrogen in the axis labels. The lines represent the linear regression line fitted (using ordinary least-squares regression) to the data points. We added the p-values from these linear regression analyses to the plots and updated the figure legend accordingly.

The analysis of current and previous contraceptive users must include age and BMI. This ensures that these two influential traits do not compromise the study (e.g., younger women are likely to be leaner and current users than older women). Also, consider the type of contraceptive as in Fig 6. Please remove women who donated blood after giving birth and if they did not restart the use of contraceptives.

Models investigating current and previous use of oral hormonal contraception did include age and BMI as covariates (described in the Material and Methods section). In addition, we added this information also to the respective paragraph in the results section (lines 523-524, section “No Long-Lasting Effect of Hormonal Contraceptives on the Plasma Proteome Observed”). As suggested by the reviewer, we repeated the analysis on the data subset of women who declared to have never been pregnant before (lines 523-524 in results section “No Long-Lasting Effect of Hormonal Contraceptives on the Plasma Proteome Observed” and new Supplementary Figure S13). Also in this analysis, no long-lasting effect of HCU was found. Regarding the different types of contraceptives, we do not have any information on what type(s) of preparations women with previous contraceptive use were taking. We can only infer this information from women currently taking contraceptives. We can therefore not assess the long-term effects of different types of preparations.

Please discuss the results of a recent preprint (<https://doi.org/10.1101/2024.08.12.607634>) in which TFF3 was highlighted (and replicated from earlier studies).

We appreciate the reviewer for bringing this new preprint to our attention, which was unavailable at the time of our manuscript submission. We have incorporated it into the discussion (paragraph 5 in the Discussion section, lines 572-576). Unfortunately, TFF3 was not measured in this study.

Are there any findings (proteins) related to the female reproductive system (as compared to the study above), or is the highly abundant proteome not built on proteins originating from female organs?

The high abundant plasma proteome mostly contains acute-phase proteins, apolipoproteins, coagulation factors, and members of the complement system, which are expressed in the liver. However, we measured sex hormone-binding globulin (SHBG), which was found to be associated with sex and hormonal contraceptive use.

Other concerns:

Many sections in the abstract and introduction don't align with the study's main objective and message. These appear as artifacts of previous versions of the manuscript. Overall, the story would be much easier to read and digest if unrelated aspects (lines 65-96) were replaced by more topic-specific information. Please remove any unnecessary and redundant technical side stories to mature the manuscript further. For example, focus on the main medical message rather than the material or method.

We appreciate the reviewer's suggestion and have made the manuscript more concise by reducing technical aspects from the introduction as well as reducing and moving technical content in the results section from the main manuscript to the supplement (i.e., Figure 5 and Table 2 providing redundant information to the written text). However, one of the main objectives of the study was also the creation and evaluation of a large-scale, high-quality proteome data set using a mass spectrometry-based setup. Thus, we kept these results, which characterize the plasma proteome of a generally healthy population and provide also important information on the circulating proteins that are being reliably quantified. We however revised the second paragraph in the "Protein Coverage and Variation in the CHRIS Cohort" results section to improve the description and reduce technical content. Further important results highlighted by previous reviewers include the unexpectedly large influence of HCU on the plasma proteome and the related need to account and adjust for this medication in any plasma proteome study as well as the comparison to other regular medication, the reduced impact of combined preparations with bioequivalent estrogens and the absence of a long-lasting effect of HCU on the plasma proteome.

A supplementary with > 100 figures seems unnecessary, and it was highly inconvenient to find the added figures that would be of value. Consider providing separate files for ease of access. Consider creating a website (e.g., Shiny app) where users can interact with the data.

We grouped the mentioned protein abundance density distributions into figures with 10 plots each and removed them from the main supplement document to increase readability and findability of figures and tables. The density distribution plots are now provided in a separate zip archive. Creating a website with individual-level data appears unfeasible due to GDPR regulations and the study's informed consent.

All figure legends are complicated and lack a short descriptive title.

We updated the figure legends and provide now more information as well as short interpretation of the data in the figures.

The demographics of the second cohort were missing. Please add and focus on females only.

The demographics of the BASE-II study subgroup are listed in Table S45, we apologize that the demographics were not better highlighted. In the revised manuscript, we now further expanded the study cohort paragraph, in the material and methods section. As described in the respective results section, the replication analysis was performed only on data from women <40 years from the BASE-II cohort.

Present global CVs, correlations, and such in density plots to provide a better overview of the data. Use IQR instead of CVs when talking about the variance in a population (= across samples), since CVs are meant to convey precision.

We added as suggested IQR (across study samples) and CV to each density plot.

Avoid the term "signal" and use "data" instead (also in supplementary).

We replaced "signal" with either "data" or "abundance" (depending on the context).

The second cohort is built on serum samples, so the title should read "Human Circulating Proteome" instead.

We thank the reviewer for pointing this out. We changed the title accordingly.

Line 548 should read: "...one of the larger..."

We replaced “largest” with “larger”.

Pasting tables into the text flow did not help; add to the end.

As suggested, we moved all tables to the end of the manuscript.

Reviewer #2 (Remarks to the Author):

After the first round of review, the authors have done major work on the manuscript. The original manuscript has improved greatly, and the authors have really taken into account the reviewers' comments. The work has also improved in readability, and it is now easier for the wider audience to read. This is great work, and now the results and discussion are more clear. Furthermore, adding the subanalysis of contraceptives with differing estrogens adds scientific value to the work. Even though I still might have formulated the discussion a bit differently, the work has improved greatly, and I think that the clinical/epidemiological value of the results is now clear enough to publish.

We thank the reviewer for the positive feedback and the constructive discussion that helped to improve our work considerably.

Response to Referees

We thank the reviewer for the constructive comments and discussion. We addressed all final minor points of the reviewer. These are listed below in italics font with our replies, shown in blue font color.

Reviewer #1 (Remarks to the Author):

The authors have done excellent work addressing all my concerns. I only have the following minor suggestions, which I think should not require another round of review and can be handled by the editorial team. Still, there were some minor artifacts from earlier versions, requiring some final fine-tuning.

1) Title: The element "Broad Population Study" is not a good fit. Two independent cohorts were used in the end. Suggest to revise in the direction of "population-based studies reveal..."

We thank the reviewer and reformulated the title in the revised manuscript accordingly.

2) Abstract: As indicated above, both cohorts should be mentioned (it's a strength). The current "Background" description of CHRIS should be part of the "Methods," and the new "Background" should describe the more general use of population cohorts to explore influential, modifiable, and non-modifiable factors in the circulating proteome that assist us to describe human health.

We rewrote this part of the abstract as suggested by the reviewer.

3) Discussion: (i) Please add the limitation of only using European-based cohorts, hence missing to determine how the discovered effects would influence the circulating proteome in other populations and ancestries. (ii) Please discuss the mechanism of OC-induced proteome changes, for example, via the liver's CYP450 system. (iv) Please dare to speculate how drug-induced liver activating could drive their observations. (v) Please refer to this previous metabolism-focussed OC study <https://doi.org/10.1093/ije/dyw147>, in which similar observations were made.

We added the requested limitation of our study and also discuss the possible involvement of enzymes from the p450 family. Finally, we cite and discuss in the revised manuscript also, as requested, the metabolomics-based publication reporting similar findings.

Overall, this is an excellent study that presents important insights into the impact of regular and common medication use in females.

We thank the reviewer for the kind words!